# Electrochemically induced crystalline-to-amorphization transformation in sodium samarium silicate solid electrolyte for long-lasting sodium metal batteries

Ge Sun[1,4], Chenjie Lou[2,4], Boqian Yi[1], Wanqing Jia[1], Zhixuan Wei[1], Shiyu Yao [1]✉, Ziheng Lu [3] ✉, Gang Chen[1], Zexiang Shen [1], Mingxue Tang [2] ✉ & Fei Du [1]✉

Exploiting solid electrolyte (SE) materials with high ionic conductivity, good interfacial compatibility, and conformal contact with electrodes is essential for solid-state sodium metal batteries (SSBs). Here we report a crystalline $Na_5SmSi_4O_{12}$ SE which features high room-temperature ionic conductivity of $2.9 \times 10^{-3}$ S cm$^{-1}$ and a low activation energy of 0.15 eV. All-solid-state symmetric cell with $Na_5SmSi_4O_{12}$ delivers excellent cycling life over 800 h at 0.15 mA h cm$^{-2}$ and a high critical current density of 1.4 mA cm$^{-2}$. Such excellent electrochemical performance is attributed to an electrochemically induced in-situ crystalline-to-amorphous (CTA) transformation propagating from the interface to the bulk during repeated deposition and stripping of sodium, which leads to faster ionic transport and superior interfacial properties. Impressively, the Na|$Na_5SmSi_4O_{12}$|$Na_3V_2(PO_4)_3$ sodium metal batteries achieve a remarkable cycling performance over 4000 cycles (6 months) with no capacity loss. These results not only identify $Na_5SmSi_4O_{12}$ as a promising SE but also emphasize the potential of the CTA transition as a promising mechanism towards long-lasting SSBs.

Solid-state batteries (SSBs) are expected to provide key improvements over today's rechargeable batteries owing to the inherent merits of solid electrolytes (SEs) such as high safety, long-lasting life, and high energy density[1,2]. Among them, sodium-based SSBs are drawing ever-increasing interest because of the abundant resource in nature, beneficial to cost saving and sustainability[3,4]. Despite the progress of sodium-based SSBs in the past few years, it is still a significant challenge to exploit low-cost and facile synthesized SEs with high ionic conductivity, excellent mechanical and chemical stability. Moreover, the poor wetting of the solid-solid interface with sluggish interfacial kinetics is a big hurdle to future Na-SSBs development[5,6].

Na-based inorganic SEs can be divided into two categories, e.g., sulfides like $Na_3SbS_4$, $Na_{11}Sn_2PS_{12}$, $Na_7P_3S_{11}$ and oxides including Na-β"-$Al_2O_3$, NASICON type[7]. Sulfide electrolytes feature higher ionic conductivity and better ductility than the oxides SEs[8,9]. However, their chemical instability against air and narrow electrochemical windows are likely to induce complex side reactions, leading to shortened cycling lives[10,11]. In contrast, NASICON-type oxides could deliver high thermal and chemical stability, and low thermal expansion[12,13]. Nevertheless, large grain boundary resistance and harsh synthesis conditions are critically challenging for their application[14,15]. In 1978, Shannon et al. first reported the synthesis of a new inorganic material $Na_5MSi_4O_{12}$ and

[1]Key Laboratory of Physics and Technology for Advanced Batteries (Ministry of Education), State Key Laboratory of Superhard Materials, College of Physics, Jilin University, 130012 Changchun, China. [2]Center for High Pressure Science and Technology Advanced Research (HPSTAR), 100193 Beijing, China. [3]Department of Materials Science & Metallurgy, University of Cambridge, 27 Charles Babbage Road, Cambridge CB3 0FS, UK. [4]These authors contributed equally: Ge Sun, Chenjie Lou. ✉e-mail: yaoshiyu@jlu.edu.cn; zluag@connect.ust.hk; mingxue.tang@hpstar.ac.cn; dufei@jlu.edu.cn

                                                 

found that the ionic conductivity gradually increases with the increase of $M^{3+}$ ionic radius with ionic conductivity of $10^{-1}$ S cm$^{-1}$ at 200 °C[16]. Recently, our group achieved a room-temperature ionic conductivity of $1.59 \times 10^{-3}$ S cm$^{-1}$ for $Na_5YSi_4O_{12}$ via optimization of the synthesis condition[17]. In comparison with the NASICON-type materials, $Na_5YSi_4O_{12}$ can be synthesized at a lower temperature which is beneficial to cost- and energy-saving. More impressively, the stable structure provides sufficient freedom of materials optimization and design by substituting the Y sites by for instance, In, Sc, and the rare earth Lu-Sm, to further enhance the ionic conductivity and understand the ion-conducting behavior of this class of materials[18].

Besides the difficulty in the SEs exploration, another challenge of developing Na-SSBs lies in the electrode–electrolyte interfaces, in the mechanics throughout the cell, and in processing at scale[19]. To begin with, the poor ductility and crystallographic orientation-dependent ionic transport properties of oxide SEs likely induce a large interfacial resistance and sluggish kinetics[20,21]. In addition, during the continuous dissolution of sodium, the formation of pores at the sodium metal anode interface will further worsen the interfacial physical contact and generate unavoidable surface defects that could disturb sodium-ion flux and work as the nucleation center, leading to rapid dendrite nucleation and growth[22,23]. Moreover, most SEs are intrinsically thermodynamically unstable against sodium metal, which induces degradation and forms mixed conducting interphases, further accelerating the dendrite propagation[24,25]. To address these issues, many approaches are employed to fabricate an artificial layer on the surface of sodium metal to improve sodium wetting, chemical stability, and thus reduce interfacial impedance, such as $TiO_2$[26], $SnS_2$[27], $AlF_3$[28], and polymer with intrinsic nanoporosity (PIN)[29]. However, they face practical limitations, such as complex synthesis procedures and difficulty in controlling the thickness and achieving acceptable adhesion. Even worse, the artificial interface layer is likely to introduce additional interfacial issues with bulk SEs, such as unexpected phase transition, uneven ion flux distribution, and electrostatic potential drop and formation of "space-charge layer," seriously limiting the ion transport and reducing the cycle life of SSBs[30].

Herein, we report the synthesis and electrochemical properties of $Na_5SmSi_4O_{12}$, which has the highest room-temperature ionic conductivity of $2.9 \times 10^{-3}$ S cm$^{-1}$ among the $Na_5MSi_4O_{12}$ family that has been reported. Interestingly, we observe an electrochemically induced crystalline-to-amorphous (CTA) transformation of $Na_5SmSi_4O_{12}$ SE during repeated deposition and stripping of Na. This CTA transition is attributed to the lattice stress generated upon Na$^+$ transportation rather than phase transformation due to chemical instability. When applied in a Li symmetric cell with the same cell configuration (Li|Na$_5$SmSi$_4$O$_{12}$|Li), this CTA process is speeded up because of the mismatch between Li$^+$ and Na$^+$ ionic radius, which further improves the selectivity of the Li SSBs. Beneficial from the enhanced mechanical properties, decreased ion mobility activation energy, and lower interfacial energy of amorphous material and interface than crystalline $Na_5SmSi_4O_{12}$, symmetric Na cells deliver a low overpotential of ~26 mV and stable cycling performance over 800 h at 0.15 mA cm$^{-2}$. Moreover, the amorphous $Na_5SmSi_4O_{12}$ facilitates intimate contact of SE with Na metal and brings essentially improved critical current density (CCD) of 1.4 mA cm$^{-2}$ in comparison with the initial crystalline stage (0.6 mA cm$^{-2}$). By virtue of the decreased resistance of sodium metal anode, the assembled quasi-solid-state Na|Na$_5$SmSi$_4$O$_{12}$|Na$_3$V$_2$(PO$_4$)$_3$ cell demonstrates an ultra-long cycle lives over 4000 cycles with ~100% Coulombic efficiency and capacity retention, indicative of the promising application of $Na_5SmSi_4O_{12}$ SE in future large-scale energy storage.

## Results

### Synthesis and characterization of crystalline $Na_5SmSi_4O_{12}$

Rhombohedral-prismatic crystalline $Na_5SmSi_4O_{12}$ was successfully synthesized via two-step solid-state reaction. Detailed synthesis processes are given in the experimental sections (Supplementary Figs. 1 and 2).

Note that the sintering temperature of $Na_5SmSi_4O_{12}$ is lower than other oxide SEs (Supplementary Table 1), such as $Na_5YSi_4O_{12}$, Na-β"-Al$_2$O$_3$, $Na_3Zr_2Si_2PO_{12}$ and its derivatives, etc., good for the cost- and energy-saving. As shown in Fig. 1a, X-ray diffraction (XRD) was taken to identify the structural property of $Na_5SmSi_4O_{12}$, and the Rietveld refined parameters are listed in Supplementary Table 2. All the diffraction peaks can be indexed into a rhombohedral system with space group $R\overline{3}c$, and the lattice parameters are calculated as $a = b = 22.1461$ Å, $c = 12.6886$ Å. Energy dispersive X-ray spectroscopy (Supplementary Fig. 3) suggests all the elements of Na, Sm, Si, and O are uniformly dispersed in the as-prepared $Na_5SmSi_4O_{12}$.

The ionic conductivity of $Na_5SmSi_4O_{12}$ was then studied by the alternating current (AC) impedance spectra and the Nyquist plot at room temperature is as presented in Supplementary Fig. 4a. Via fitting the bulk and grain boundary resistance, the total ionic conductivity of $Na_5SmSi_4O_{12}$ is calculated as $2.9 \times 10^{-3}$ S cm$^{-1}$, among the highest values in the existing ceramic electrolytes (Supplementary Table 1). Activation energy ($E_a$) of $Na_5SmSi_4O_{12}$ was further evaluated by fitting the Arrhenius plot (Supplementary Fig. 4b). $E_a$ is calculated as small as 0.15 eV, which is smaller than that of $Na_5YSi_4O_{12}$, indicative of a fast ionic transportation ability. In addition, the electronic conductivity of $Na_5SmSi_4O_{12}$ was measured as about $5.8 \times 10^{-10}$ S cm$^{-1}$ via a direct current (DC) polarization measurement (Supplementary Fig. 5). The intrinsic electronic insulation can effectively reduce the self-discharge of batteries and suppress dendrite growth, enabling $Na_5SmSi_4O_{12}$ as a good candidate for sodium-based SSBs[31]. Besides the merits of high ionic and low electronic conductivities, $Na_5SmSi_4O_{12}$ demonstrates superior moisture stability, whose XRD pattern shows no change after soaking in deionized water for 48 h (Supplementary Fig. 6) or exposed to air for 45 days (Supplementary Fig. 7), showing great potential in future industrial application.

$Na_5SmSi_4O_{12}$ demonstrates similar ionic transportation processes as $Na_5YSi_4O_{12}$[17] and detailed calculation results are listed in Supplementary Fig. 8a–d. To be specific, the conductivity is contributed by a percolating conduction pathway in the mobile region, whereas the rest of Na ions serve as a pillar to hold the structure together. Interestingly, along the ion conduction pathway, three distinctive sites are revealed where Na ions can stably sit, denoted as sites A, B, and C. These sites are connected to each other via the zigzag-like channel as observed from the molecular dynamics simulation trajectory. Furthermore, the diffusion barrier of Na ions between these sites is found to be ~0.3 eV, which is relatively low in comparison with other reported solid ion conductors[32,33]. However, such a value is larger than the activation energy from experiment. We assign such a deviation to the concerted Na hopping behavior. The ~0.3 eV barrier was calculated by assuming a vacancy-mediated uncorrelated conduction mechanism, while the Na concentration is relatively high and concerted motion is favored. The case of concerted motion is further estimated by assuming a two-ion correlated hopping mechanism, as shown in Fig. 1b. According to such a mechanism, the diffusion barrier along the same path dropped to ~0.19 eV, close to the experimental value.

Furthermore, solid-state $^{23}$Na nuclear magnetic resonance (NMR) measurements were carried out to reveal the atomic local structure and dynamics mobility[34–36]. As exhibited in Fig. 1c, the pristine $Na_5SmSi_4O_{12}$ conductor shows multiple $^{23}$Na resonances, which can be further deconvoluted into six types of signals. Via the aid of crystal structure and the proportion of sodium, the peak at 8.8 ppm is assigned to the mobile sodium at Na5 site, the resonance at 4.5 and 1.6 ppm are attributed to Na1 and Na3, the signal at −16.8 ppm is from Na4, and the rest peaks at −23.4 and −28.6 ppm are assigned to sodium at Na2 and Na6 sites, respectively. The simulated details are listed in Supplementary Table 3, from which the proportion of each Na atom agrees with the theoretical results. To obtain more information about Na$^+$ dynamics properties, NMR spectra at different temperatures were performed. As displayed in Supplementary Fig. 9, the $Na_5SmSi_4O_{12}$

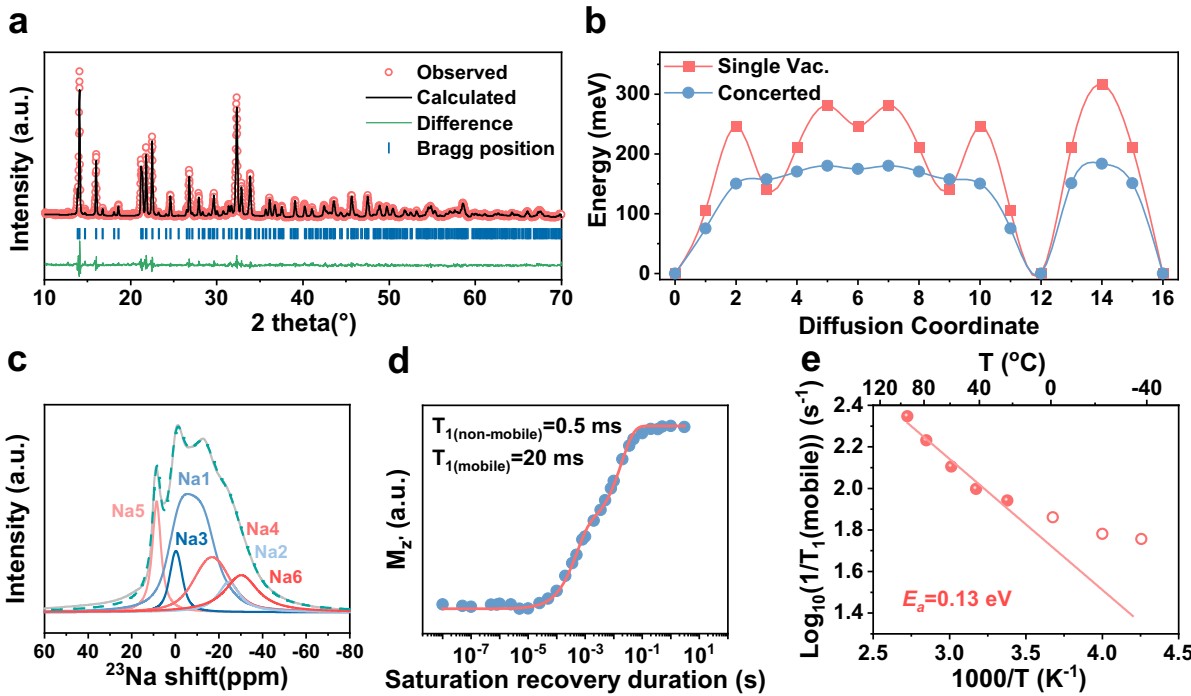

**Fig. 1 | Crystal structure and sodium-ion conduction characteristic of crystal-line Na₅SmSi₄O₁₂. a** Rietveld refinement based on the powder XRD. **b** Minimum potential energy path along Na⁺ diffusion route in crystalline Na₅SmSi₄O₁₂. **c** Solid-state ²³Na NMR spectrum and its simulation for the crystalline Na₅SmSi₄O₁₂. The gray line is experimental data and the green-dashed line is the sum of simulation. **d** Saturation recovery fitting curve for the data obtained at room temperature. **e** Temperature dependence of ²³Na NMR relaxation rate as a function of temperature in K⁻¹. The solid line is the fit according to Eq. (1). The derivation of the data is not used for the fit.

show similar spectra upon increasing temperature from 213 to 367 K, except a decrease in signal due to Boltzmann distribution. No obvious change is observed for spectral configuration, possibly attributed to the stable structure caused by the non-mobile Na ions at Na1, Na2, and Na3 sites. Therefore, we turn to a more temperature-sensitive parameter, spin-lattice relaxation (SLR) time ($T_1$). Since the strong quadrupole interaction is observed for ²³Na NMR spectrum in our system, the saturation recovery technique is employed to determine $T_1$ values[37]. Via fitting the spectral intensity vs. saturation recovery period measured at room temperature (Fig. 1d), two relaxation time values of 0.5 and 20 ms are obtained for the non-mobile and mobile Na ions, respectively[38]. Here, the data analysis is simplified to non-mobile and mobile for convenience. According to Eq. (1), the fitting of the relaxation times as a function of temperature yields an activation energy $E_a \approx 0.13$ eV for the mobile Na⁺, which is in good agreement with the results from AC impedance (0.15 eV).

$$R_1 = 1/T_1 \propto \omega_0^\beta \exp[-E_a/(kT)] \tag{1}$$

Where $R_1$ is NMR spin-lattice relaxation rate, the reciprocal of $T_1$, $\omega_0$ is resonance frequency, $\beta$ is modified exponent, $E_a$ is activation energy, $k$ is Boltzmann constant, $T$ is the absolute temperature in K. Note that the derivate data points marked by hollow circle, as displayed in Fig. 1e, were excluded from the fits since the $R_1$ rates ($1/T_1$) recorded at low temperature range are mainly governed by non-diffusive background effects, such as lattice vibrations or coupling by paramagnetic impurities[39–41].

**Electrochemical performance and crystalline-to-amorphous (CTA) transition of Na₅SmSi₄O₁₂ solid electrolyte**

To further evaluate the chemical and electrochemical stability of SE against Na metal, a symmetric all-solid-state Na cell using Na₅SmSi₄O₁₂ as the electrolyte was fabricated and tested by the repeatedly

galvanostatic stripping and plating at different current densities. The charge/discharge profiles of Na|Na₅SmSi₄O₁₂|Na without any interfacial modification were recorded at a current density ranging from 0.05 to 0.15 mA cm⁻² with 120 min per cycle (Fig. 2a). The cell displays stable and long cyclic performance which maintains an overvoltage of ~26 mV at 0.15 mA cm⁻² with negligible fluctuations over 800 h, indicating sodium dendrite-free plating/stripping and excellent kinetic stability of the Na₅SmSi₄O₁₂ against Na metal. Afterwards, electrochemical impedance was collected from the symmetric cell after cycling 150, 200 and 300 h to reveal the changes in the internal resistance, as illustrated in Fig. 2b and Supplementary Table 4. The resistance of SE (both $R_b$ and $R_{GB}$) remains nearly unchanged as the sodium plating/stripping proceeds. In contrast, the interfacial resistance ($R_{int}$) between sodium metal and SE decreases at the initial few cycles and then stabilizes at a relatively low value, demonstrating excellent compatibility between Na₅SmSi₄O₁₂ and sodium metal. This phenomenon is quite different from most of the reported SEs without surface modification, whose $R_{int}$ increases gradually with increasing cycling times as the result of the excessive internal resistance and failure of the cells due to interfacial side reactions or the formation of pores[25,28,42].

Scanning electron microscopy (SEM) images of the electrodes at different plating/stripping stages are shown in Fig. 2c, d, which demonstrates an interesting tendency of gradually vanishing gap at the interface between Na and Na₅SmSi₄O₁₂ after cycling and a compact interfacial contact was created. XRD pattern after cycling 200 h suggests a transformation from the crystalline into the amorphous state on the surface of Na₅SmSi₄O₁₂ SE (Fig. 2e). To further assess the amorphous depth of Na₅SmSi₄O₁₂, XRD was utilized to monitor the structural changes at various polishing depths, as depicted in Supplementary Fig. 10. The analysis reveals that the SE pellet, after undergoing 100 h of cycling at 0.15 mA cm⁻², exhibits no reflections, indicating complete amorphization near the surface of the Na₅SmSi₄O₁₂ SE. As the polishing depths increased, a limited number

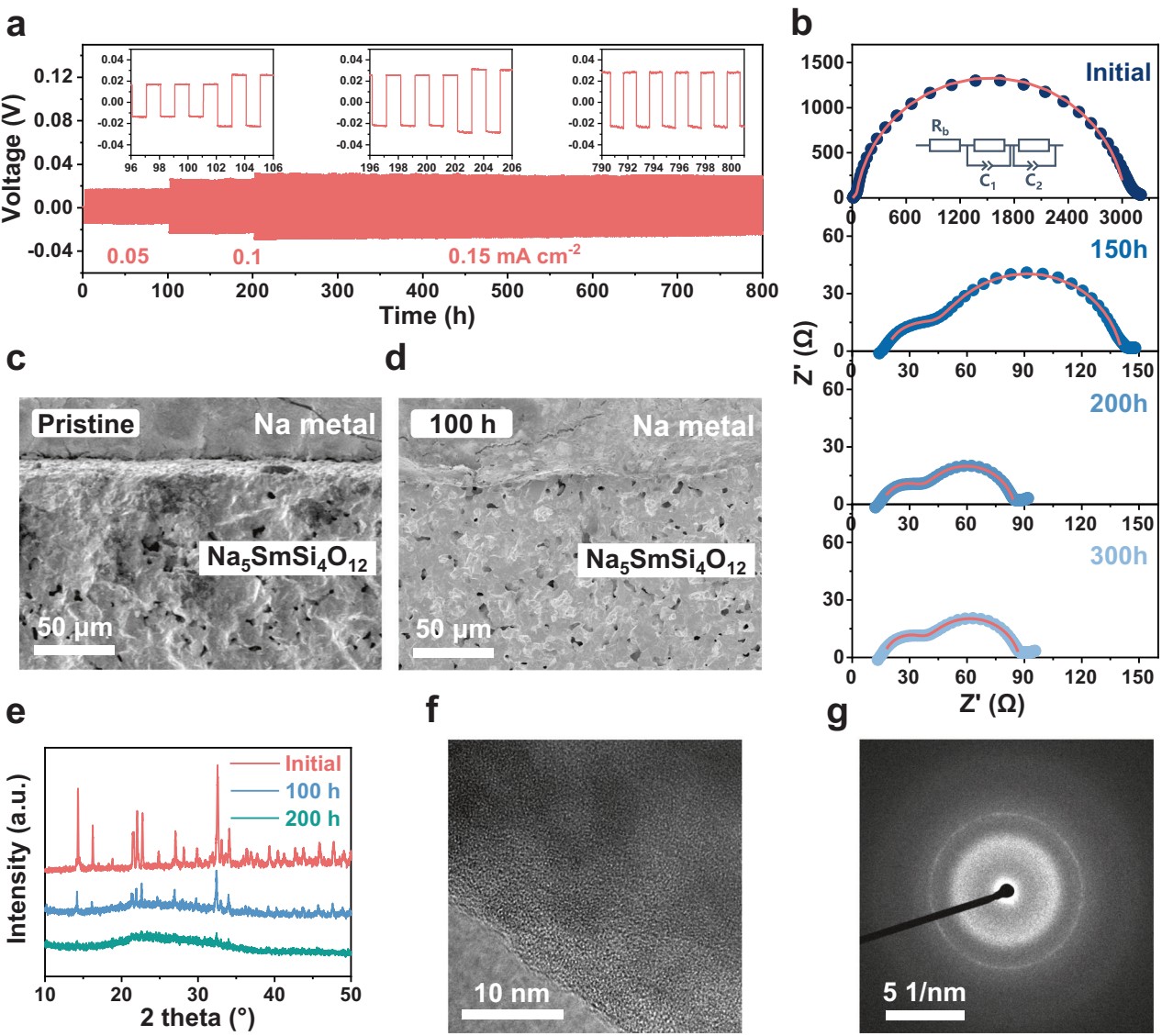

**Fig. 2 | Characterization of interfacial evolution cycling in Na|Na$_5$SmSi$_4$O$_{12}$|Na symmetric cells. a** Cycling performance of Na|Na$_5$SmSi$_4$O$_{12}$|Na at room temperature. **b** Nyquist plots of the SE-based symmetric cell after cycling for different times. Cross-sectional SEM images of the Na metal/Na$_5$SmSi$_4$O$_{12}$ interfaces: **c** pristine and **d** 100 h cycling. **e** XRD profiles of Na$_5$SmSi$_4$O$_{12}$ after cycling for different times. **f** HRTEM and **g** SAED patterns of Na$_5$SmSi$_4$O$_{12}$ after cycling.

of weak reflections of Na$_5$SmSi$_4$O$_{12}$ emerge, which suggests a gradual amorphization process from the surface to the bulk. Upon polishing to a depth of 0.15 mm, sharp reflections appear which means the crystalline SE may constitute the majority of the volumetric ratio within the entire Na$_5$SmSi$_4$O$_{12}$ chip. To quantify the weight fraction of amorphization in an SE sample and by what kinetics, the internal standard method was employed to further assess the CTA transformation process, by mixing CeO$_2$ with the Na$_5$SmSi$_4$O$_{12}$ SE at different plating/stripping stages. The Rietveld refinements and the corresponding results are summarized in Supplementary Figs. 11 and 12, respectively, which clearly reveal the weight fraction of amorphization is strongly related to both the cycling time and current density. With increasing the cycling time, a higher weight fraction of the amorphous phase can be realized and a higher applied current density can accelerate the amorphization process. Additionally, the high resolution transmission electron microscope (HRTEM) and selected area electron diffraction (SAED) measurements were undertaken to further confirm the CTA transition before (Supplementary Fig. 13) and after cycling (Fig. 2f, g). There are no observed lattice fringe and diffraction spot for the cycled SE sample, confirming the electrochemistry-induced CTA transition.

Because of the amorphous state, it is difficult to determine the possible local coordination based on the XRD measurement. Therefore, Raman spectra were then used to examine the short-range vibration changes. As displayed in Supplementary Fig. 14, crystalline Na$_5$SmSi$_4$O$_{12}$ exhibits seven vibration peaks[43]: the bands in the 900–1100 cm$^{-1}$ region are assigned to Si-O stretching vibrations, the symmetrical band at ~624 cm$^{-1}$ is corresponding to O-Si-O bending mode. The low-frequency bands (<550 cm$^{-1}$) are attributed to Sm-O and Na-O bond vibrations in their polyhedral. Though the cycled Na$_5$SmSi$_4$O$_{12}$ SE loses its long-term ordering (Fig. 2e–g), it maintains nearly all the vibrational peaks that confirm no chemical reaction between the interface of SE and Na metal. The disappearance of the peak at 1041 cm$^{-1}$ might be related to the damage of the fracture of Si-O bond by the amorphous transition. Furthermore, X-ray photoelectron spectroscopy (XPS) suggests that there are no changes in the Sm 3$d$, Na 1$s$, and Si 2$p$ XPS spectra after cycling, indicative of no redox reaction occurred between sodium metal and Na$_5$SmSi$_4$O$_{12}$ (Supplementary Fig. 15). Thus, it can be concluded that Na$_5$SmSi$_4$O$_{12}$ SE demonstrates an interesting CTA transition with excellent electrochemical stability that enables it as the ideal SE for sodium-based SSBs.

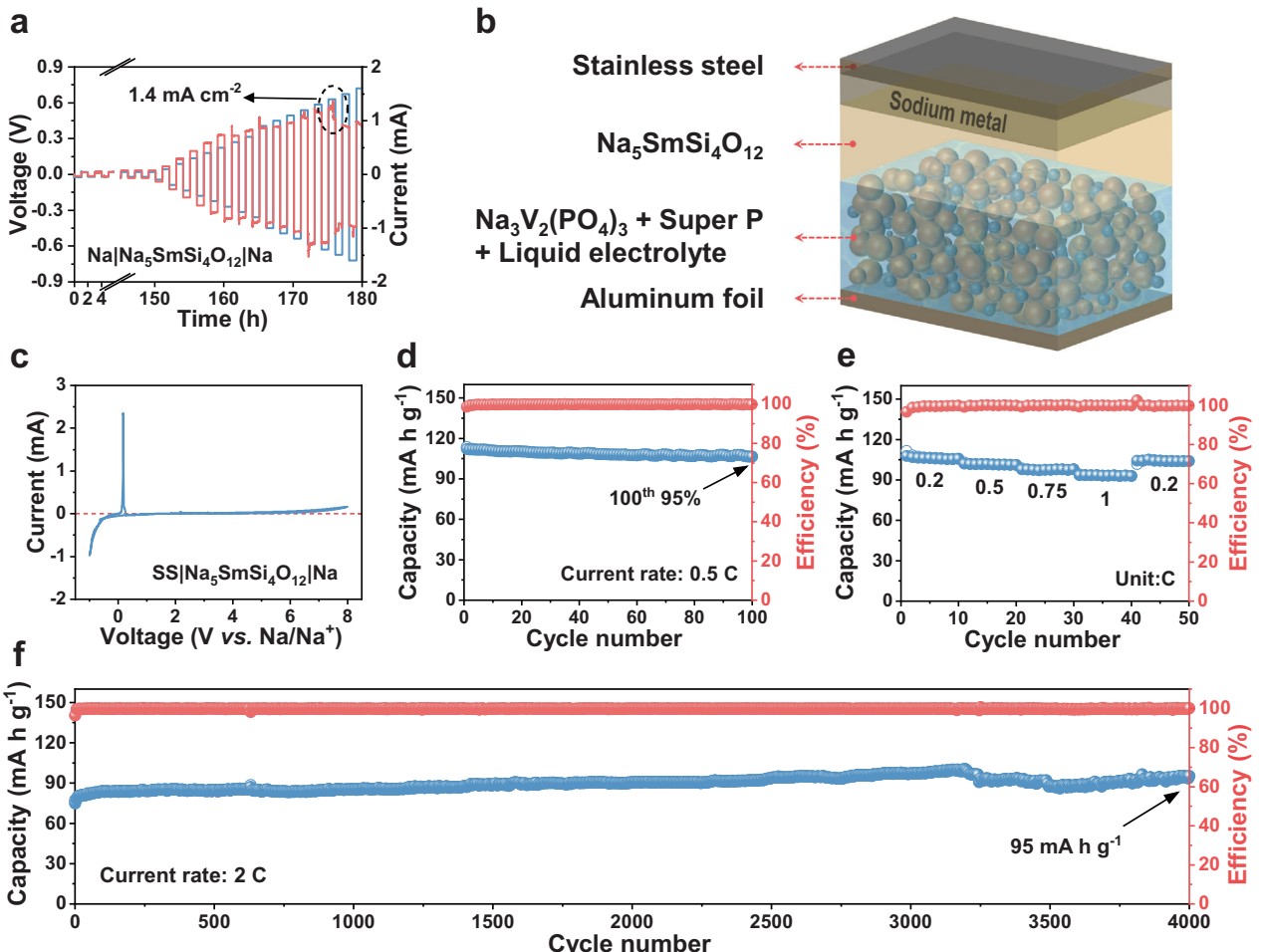

**Fig. 3 | The electrochemical performance of solid-state batteries. a** Potential response of Na|Na$_5$SmSi$_4$O$_{12}$|Na cell during the CCD measurement with a low area capacity of 0.05 mA h cm$^{-2}$ is applied for 150 h in advance. **b** Schematic illustration of Na|Na$_5$SmSi$_4$O$_{12}$|Na$_3$V$_2$(PO$_4$)$_3$. **c** CV curve of SS|Na$_5$SmSi$_4$O$_{12}$|Na cell at a scanning speed of 5 mV s$^{-1}$. **d** Cycling performance at 0.5 C-rate (1 C corresponds to 118 mA g$^{-1}$). **e** Rate capability at 0.2, 0.5, 0.75, and 1 C-rate. **f** Long-term cycle life at 2 C-rate.

## Electrochemical performances of solid-state batteries

Critical current density (CCD) and long-term cycling performance of Na| Na$_5$SmSi$_4$O$_{12}$|Na with a long single deposition time were further measured to evaluate the capability of amorphous materials and interface in suppressing dendrite growth. As displayed in Fig. 3a and Supplementary Fig. 16, the amorphous Na$_5$SmSi$_4$O$_{12}$ facilitates intimate contact of SE with Na metal and brings essentially improved CCD of 1.4 mA cm$^{-2}$ in comparison with the initial crystalline stage (0.4 mA cm$^{-2}$). This behavior is mainly because the uneven metal deposition leads to rapid growth of sodium dendrites along the crystalline Na$_5$SmSi$_4$O$_{12}$ grain boundary with a large deposition current and hence results in the short circuit rapidly. The result is also much higher than most reported values based on oxide SEs without and with SE/Na metal anode interfacial modification (Supplementary Table 6), indicating the superiority of amorphous SE interfaces. In addition, as shown in Supplementary Fig. 17a, under a large area capacity, the overpotential increases during the cycling and drops suddenly less than 25 h, which suggests that crystalline Na$_5$SmSi$_4$O$_{12}$ can be easily penetrated by sodium dendrites. By contrast, if a low area capacity of 0.05 mA h cm$^{-2}$ is applied in advance to make the electrolyte transition to an amorphous state, the Na|amorphous Na$_5$SmSi$_4$O$_{12}$|Na cell displays a stable and long cycling performance which maintains the overvoltage at around 20 mV over 500 h (Supplementary Fig. 17b). All these results indicate the strong capability of amorphous Na$_5$SmSi$_4$O$_{12}$ in suppressing Na dendrite formation.

To further emphasize the superiority of amorphous interface and bulk materials, a Na|Na$_5$SmSi$_4$O$_{12}$|Na$_3$V$_2$(PO$_4$)$_3$ (NVP) SSB was constructed and evaluated at room temperature, as illustrated in the schematic figure (Fig. 3b). The cyclic voltammetry (CV) curve of stainless steel (SS)|Na$_5$SmSi$_4$O$_{12}$|Na cell in Fig. 3c shows that Na$_5$SmSi$_4$O$_{12}$ possesses a wide electrochemical stability window more than 5 V, which is high enough to ensure that the electrolyte does not undergo phase transition within the working voltage range. As shown in Supplementary Fig. 18a, quite flat charge–discharge voltage profiles at 2.3–3.9 V with an initial discharge capacity of 112 mA h g$^{-1}$ were observed, which matches well with the characteristic NVP redox plateaus in liquid electrolyte (Supplementary Fig. 18b). In addition, the high initial Coulombic efficiency of 99% indicates that there is no irreversible side reaction between Na$_5$SmSi$_4$O$_{12}$ and Na anode or NVP cathode. Meanwhile, an excellent cycling performance is achieved with high-capacity retention of 95% after 100 cycles (Fig. 3d). Furthermore, specific capacities of 102, 98, and 93 mA h g$^{-1}$ can be obtained at 0.5, 0.75, and 1 C-rates, respectively, suggestive of a superior rate capability (Fig. 3e). Finally, the long-term cycling stability was estimated at a current rate of 2 C (Fig. 3f). Impressively, there is no obvious capacity loss during the repeated 4000 cycles (6 months), demonstrating superiority in the state-of-the-art solid-state full cell in Supplementary Table 7. All these features verify the unique interface properties between Na$_5$SmSi$_4$O$_{12}$ and sodium, which can not only enable

sufficient contact with sodium metal, but also inhibit the side reaction and the dendrite growth during cycle process.

## Discussion

### Cell stabilization by CTA transition

According to the above-mentioned results, the electrochemical-induced CTA transition plays a key role in stabilizing the interfacial properties, suppressing the dendrite formation, and thus increasing the long-term stability and high-rate capability. This superiority of the amorphous stage from the interface to the bulk SE material can be understood in terms of the following two aspects. Firstly, the ionic conductivity of amorphous bulk material was improved. The pathway and the ion transport properties of amorphous $Na_5SmSi_4O_{12}$ were investigated in detail by solid-state NMR. Figure 4a shows the $^{23}Na$ NMR spectra before and after metallic Na cycling by using crystalline $Na_5SmSi_4O_{12}$ as an electrolyte. The $^{23}Na$ NMR spectrum did not change significantly before and after cycling, except changes in the mobile ions (Na4, Na5, and Na6), indicative of their slight redistribution upon cycling. Nevertheless, NMR results demonstrate the non-mobile and mobile segments for Na migration within a stable structure. In addition, spin-lattice relaxation time $T_1$ of $^{23}Na$ was measured at different temperatures for amorphous $Na_5SmSi_4O_{12}$, as shown in Fig. 4b and Supplementary Fig. 19. The activation energy of the mobile $Na^+$ of amorphous $Na_5SmSi_4O_{12}$ is calculated as 0.07 eV lower than that of the pristine crystalline state (0.13 eV in Fig. 1k). The decrease in the active energy strongly suggests an enhanced $Na^+$ hopping ability for the

amorphous $Na_5SmSi_4O_{12}$ with higher ionic conductivity. Secondly, the interfacial issues, such as high interfacial resistance and metal dendrite growth, are strongly alleviated. The schematic illustration of sodium deposition is provided in Fig. 4d. A locally solid−solid contact induces a heterogeneous sodium-ion flux, whereas a tight interfacial contact of amorphous $Na_5SmSi_4O_{12}$ lead to uniform deposition of sodium. As mentioned above in Fig. 2b, there is obvious decrease in the $R_{int}$ upon cycling at different plating/stripping stages. The interesting phenomenon can be attributed to compact interfacial contact between Na metal and electrolyte due to the increased contact area (Fig. 2c, d). Furthermore, the isotropic surface of amorphous material could enhance the wettability with significantly reduced interface resistance and eliminates the influence of crystallographic orientation-dependent ionic transport since the interfacial energy of amorphous $Na_5SmSi_4O_{12}$ is calculated as 0.33 J m$^{-2}$ with sodium, lower than the crystalline $Na_5SmSi_4O_{12}$ (0.56 J m$^{-2}$) (Supplementary Fig. 20). In addition, the amorphous-$Na_5SmSi_4O_{12}$ deliver the high mechanical strength, beneficial to inhibiting the dendrite growth. As shown in Fig. 4c, nanoindentation technique was employed to evaluate the Young's modulus $E$ and hardness $H$[44,45]. As for the amorphous sample, the $E$ and $H$ are calculated to be ~79.9 and ~3.8 GPa, respectively, higher than the crystalline $Na_5SmSi_4O_{12}$ (~72.6 and ~2.8 GPa). In summary, the intrinsic microstructural and compositional homogeneity, as well as low electronic conductivity, alleviate the potential fluctuations at local positions in the SE and suppress sodium propagation and penetration into amorphous $Na_5SmSi_4O_{12}$.

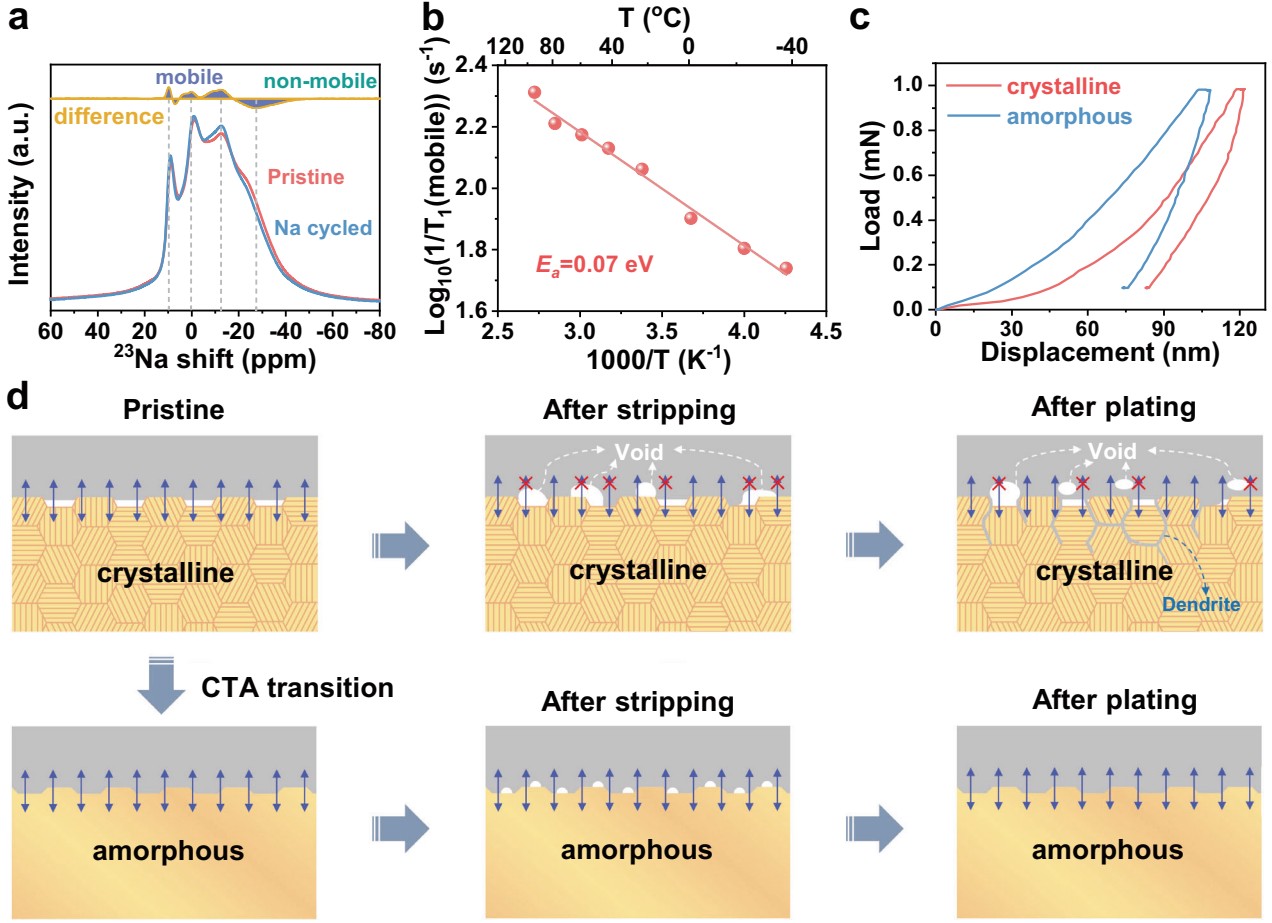

**Fig. 4 | The superiority of amorphous bulk materials and interface. a** Solid-state $^{23}Na$ NMR of the pristine crystalline and the cycled amorphous $Na_5SmSi_4O_{12}$. **b** $^{23}Na$ NMR relaxation rate of Na cycled $Na_5SmSi_4O_{12}$ as a function of temperature in K$^{-1}$. **c** Nanoindentation load−displacement curves of crystalline $Na_5SmSi_4O_{12}$ and amorphous $Na_5SmSi_4O_{12}$. **d** Schematic of interface morphology evolution during sodium plating/stripping.

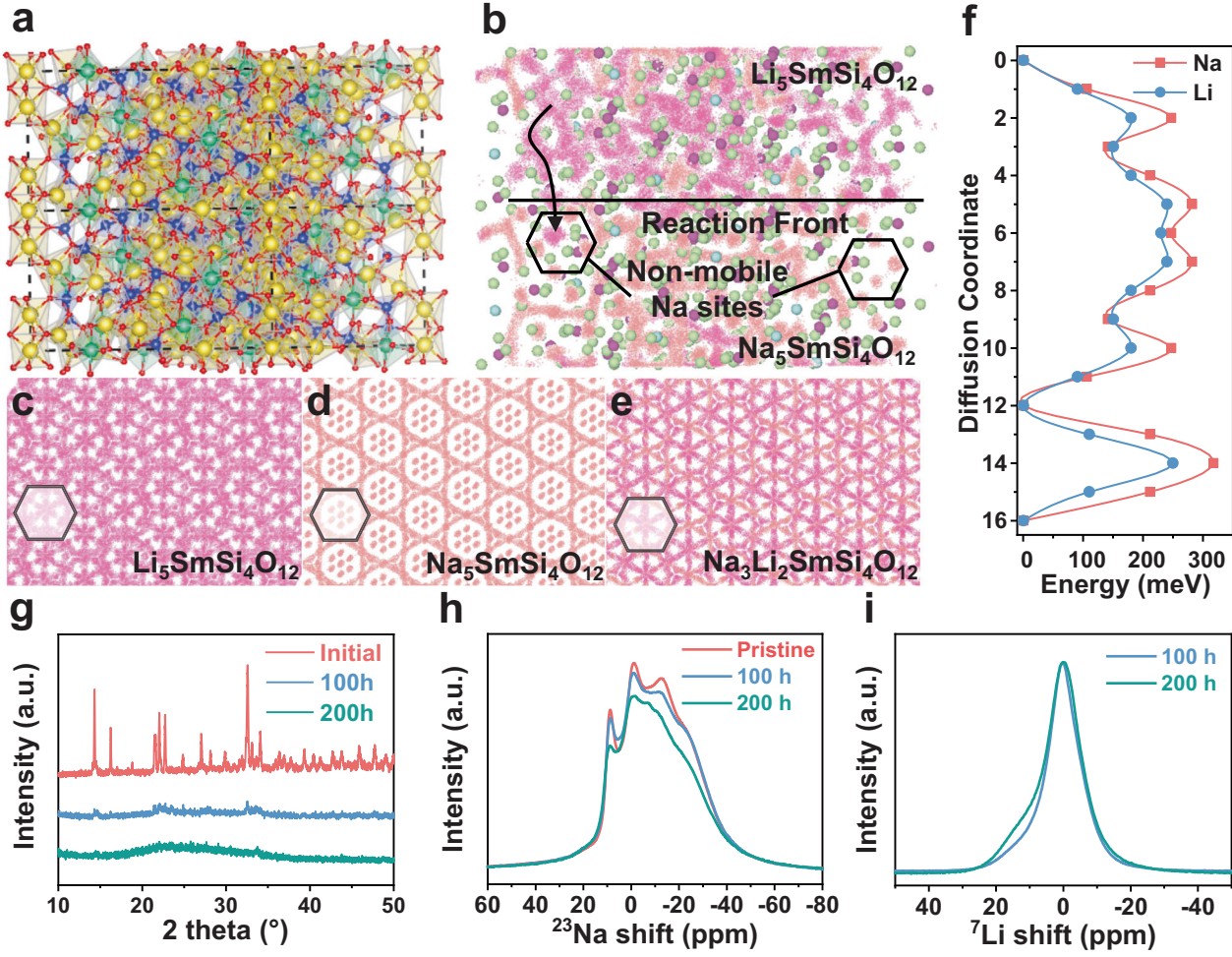

**Fig. 5 | CTA mechanism. a** Structure of the $Na_5SmSi_4O_{12}|Li_5SmSi_4O_{12}$ interface model. **b** Molecular dynamics trajectories of the interface model. Molecular dynamics simulation trajectories of crystalline **c** $Li_5SmSi_4O_{12}$ **d** $Na_5SmSi_4O_{12}$, and **e** $Na_3Li_2SmSi_4O_{12}$. **f** Minimum potential energy path along Li diffusion route in crystalline $Na_3Li_2SmSi_4O_{12}$. **g** XRD profiles of $Na_{5-x}Li_xSmSi_4O_{12}$ after cycling for different times. Solid-state **h** $^{23}Na$ and **i** $^7Li$ NMR of the $Na_5SmSi_4O_{12}$ with different cycling times.

## Mechanism of CTA transition

Electrochemically-induced solid-state amorphization (SSA) transformations are popular in the alloying type anode, like the transition from nano Si into amorphous Li-Si phases after electrochemical lithiation[46,47]. Nevertheless, this kind of SSA process usually undergoes a chemical phase transition, which is disastrous for SEs. To the best of our knowledge, it is the first observation about the electrochemically induced CTA transformation in oxide SEs during alkali metal plating/stripping without chemical phase transition. Generally, the SSA transformation depends on the thermodynamic driving force (pressure, defects, internal stress, etc.), and the existence of kinetic constraints (mainly because of the low experimental temperature) that prevent the formation of full equilibrium crystalline phases[48–50]. For example, when ion conduction is non-homogeneous, the local variance may lead to large local stress, which could in turn lead to the collapse of the structure. Through cycling, SSA transformation may occur. As presented in the schematic crystal structure of $Na_5SmSi_4O_{12}$ (Supplementary Fig. 21a), vacancies are observed at Na4, Na5 and Na6 sites, which may accommodate additional Na ions' insertion and thus induce local variance. This possible insertion can be confirmed by the presence of the initial discharge capacity of $Na_5SmSi_4O_{12}$ (Supplementary Fig. 21b) and an increase in the Na content after cycling (energy dispersive spectroscopy mapping, Supplementary Table 8). As the local stress accumulates, the internal strain field leads to the CTA

transformation. In this case, we calculate the formation energies of the crystalline and the amorphous phases sampled from quenched melts. As shown in Supplementary Fig. 22, the crystalline phases are thermodynamically favored with an energy ~60 meV atom$^{-1}$ lower than the amorphous structures. This indicates that, besides the thermodynamic force, kinetics may also play a critical role in the SSA. One way to magnify such kinetic driving force is to apply large lattice strain on the system. In this context, we computationally substitute the Na ions to Li ions in the structures and further compute the energy difference between the crystalline and the amorphous phases. Interestingly, when the Li concentration reaches ~2/5, the energy difference significantly drops to <10 meV atom$^{-1}$. This further supports our assumption that the driving force led by the large lattice strain could be the origin of the SSA. Following the above results, a Li-Na exchange process is captured by atomistic simulations, since the ionic radius of Li$^+$ (76 pm) is smaller than Na$^+$ (102 pm) that could induce much larger lattice strain. Large-scale molecular dynamics is run on the hypothetical $Na_5SmSi_4O_{12}$/$Li_5SmSi_4O_{12}$ using a machine-learned forcefield, as shown in Fig. 5a. Interestingly, the Li ions first exchanges with the mobile Na ions at the reaction front followed by mixing with the non-mobile ones, see Fig. 5b–e. When the pillar Na ions are replaced by smaller Li ions, the structure start to collapse. During the initial exchange process, crystalline $Na_5SmSi_4O_{12}$ and $Na_{5-x}Li_xSmSi_4O_{12}$ could co-exist. We computationally evaluated how Li substitution affects the ion diffusion.

As shown in Fig. 5f, when all mobile Na ions are replaced by Li, Li diffusion barrier along the original zig-zag route dropped from -0.3 to -0.25 eV, indicating a faster diffusion kinetics, which may in turn, result in faster SSA during Li exchange. In conclusion, lattice strain induced by ion intercalation is the main driving force of amorphous transformation. Once the thermodynamic driving force is present, the kinetic hindrance at room-temperature prevents the transition back to amorphous $Na_5SmSi_4O_{12}$.

Guided by the computational results, a hybrid symmetric cell of Li|$Na_5SmSi_4O_{12}$|Li was fabricated to accelerate this SSA transition. As shown in Supplementary Fig. 23, the cell reveals uniform plating and stripping overpotential profiles with an increased current density. This phenomenon suggests that hybrid movement of $Li^+$/$Na^+$ within the bulk of $Na_5SmSi_4O_{12}$ and an effective plating/stripping of $Na^+$ at Li anode. As expected, $Na_5SmSi_4O_{12}$ exhibits a much shorter amorphous time within 100 h (Fig. 5g and Supplementary Fig. 24), confirming that the CTA transition of $Na_5SmSi_4O_{12}$ is mainly triggered by the lattice strain and speeded up because of the mismatch between $Li^+$ and $Na^+$ ionic radius. Furthermore, an obvious reflections shift and weakening can be observed in the XRD patterns after experiencing different cycling time (Supplementary Fig. 25). This shift indicates the existence of microscopic strain in the lattice, and the gradual accumulation of stress leads to the break of more bonds. Thus, with the increase of cycling time, the crystallinity of $Na_5SmSi_4O_{12}$ weakens, and the intensity of XRD peaks decreases gradually. To assess the cationic electrochemical exchange mechanism, XPS analysis of SE operating in the hybrid cell was carried out after cycling. As shown in Supplementary Fig. 26, the decrease in Na $1s$ peak and appearance of Li $1s$ peak prove that $Li^+$ can successfully replace part of $Na^+$ ions and the strong $Li^+$ mobility within rhombohedral-prismatic $Na_5SmSi_4O_{12}$. In addition, there is no observed new Raman peaks after cycling in the hybrid symmetric cell (Supplementary Fig. 27), which indicates that $Na_5SmSi_4O_{12}$ SE demonstrates an excellent thermodynamic stability with Li metal.

Figure 5h shows the $^{23}Na$ spectra of $Na_5SmSi_4O_{12}$ cycled with Li metal under different times, from which the stripped Li will exchange with Na on its way when across the electrolyte. The signals at −16.8 ppm and −23.4 ppm, being assigned to sodium at Na4 and Na6 sites, significantly weakened, reflecting the transport activity of Na4 and Na6 sodium sites during polarization. In addition, the signal at 8.8 ppm, which is assigned to sodium at Na5 site, is weakened but fluctuates in intensity at different cycle times, indicating it likely serves as 'bridge' for transporting Na/Li ions during polarization. These results further conclude that there is a possible 3D pathway of $Na_5SmSi_4O_{12}$ between Na4-Na5-Na6. Figure 5i and Supplementary Fig. 28 display the $^7Li$ NMR spectrum of $Na_{5-x}Li_xSmSi_4O_{12}$ after Li cycled different times (100 and 200 h). The $^7Li$ NMR spectrum of the electrolyte cycled for 200 h is broader than that for 100 h, indicating the growth of amorphous phase. $^6Li$ NMR spectrum after Li cycling is shown in Supplementary Fig. 29. Both $^7Li$ and $^6Li$ NMR spectra present two components, corresponding to the different mobile Na sites, such as Na4 and Na6.

In summary, dense crystalline $Na_5SmSi_4O_{12}$ was prepared and exhibits a high room-temperature conductivity of $2.9\times10^{-3}$ S cm$^{-1}$. Driven by microscopic strain in the lattice, $Na_5SmSi_4O_{12}$ undergoes an amorphous transformation during the cycling in both symmetric Na|$Na_5SmSi_4O_{12}$|Na and Li|$Na_5SmSi_4O_{12}$|Li cells. On the one hand, the increased contact area greatly reduces the interfacial resistance between sodium metal and electrolyte and promotes the homogeneous deposition of sodium. On the other hand, the amorphous $Na_5SmSi_4O_{12}$ exhibits isotropic ionic transport characteristics, which effectively eliminate ion-blocking crystallographic orientations. This property promotes a uniform distribution of current and homogeneous metal nucleation at the anode interface, further enhancing the overall performance of the solid-state sodium metal battery. Thus,

the sodium symmetrical cells manifest stable cycling performance for 800 h at 0.15 mA cm$^{-2}$@1 h and 500 h at 0.05 mA cm$^{-2}$@5 h (25 °C). Furthermore, the successful operation of Na|$Na_5SmSi_4O_{12}$|$Na_3V_2(PO_4)_3$ quasi-solid-state sodium batteries with excellent electrochemical performance further implies the superiority of $Na_5SmSi_4O_{12}$ electrolyte.

## Methods

### Materials synthesis

The $Na_5SmSi_4O_{12}$ pellets were synthesized by a solid-state sintering method using $Na_2CO_3$, $Sm_2O_3$ and $SiO_2$ as the starting materials. First, the raw materials of analytical grade were mixed by ball-milling at a milling speed constant of 600 rpm for 15 h. The mixture was dried at 80 °C for 12 h and calcined at 800 °C for 8 h. Then the powder was put into a cylindrical pressing mold with diameter of 15 mm and pressed under a pressure of 300 MPa. The pressed pellets were then sintered at 950 °C for 20 h. Finally, buff pellets were obtained after sintering.

The $Na_3V_2(PO_4)_3$ (NVP) cathode material was prepared by the sol-gel method according to our previous work (Supplementary Fig. 30)[51]. Firstly, the stoichiometric amount of $Na_2CO_3$, $NH_4VO_3$ and $NH_4H_2PO_4$ with a molar ratio of 3:4:6 was dissolved in deionized water. Secondly, 0.02 M aqueous citric acid [$HOC(COOH)(CH_2COOH)_2$] solution was added dropwise into the solution until the ratio of vanadium: citric acid equals to 2:1. Then the gel could be acquired by drying the precursor in an oven at 120 °C for 12 h. Finally, the NVP/C powder can be acquired after heat treatments in two steps, first at 350 °C for 5 h and then at 750 °C for 12 h under a nitrogen atmosphere.

### Characterization method

XRD patterns were recorded by a Bruker D8 Advance diffractometer with Cu Kα radiation and RigaKu D/max-2550 diffractometer (1.6 kW, Cu Kα radiation, $\lambda$ = 1.5406 Å), followed by Rietveld refinement using Fullprof software for the crystal structure analysis. The microscopy characteristics of the samples were investigated by Hitachi Regulus8100 FESEM, high resolution transmission electron microscope (HRTEM, talos F200X) and selected area electron diffraction (SAED). The elemental mapping was used to analyze the element distribution of the samples. XPS was carried out on a thermo scientific NEXSA spectrometer. Raman spectra were examined using a Renishaw Raman microscope (model 2000) with Ar-ion laser excitation. Prior to analysis the interface properties between Na/Li metal and $Na_5SmSi_4O_{12}$ electrolyte after cycling, emery paper was employed to remove any residual metal on the surface. All $^6Li$, $^7Li$, and $^{23}Na$ magic angle spinning (MAS) NMR experiments were acquired on Bruker 400 MHz (9.4 T) magnets with AVANCE NEO consoles using Bruker 3.2 mm HXY MAS probe. The samples were filled into rotors inside Argon glove box. The Larmor frequencies for $^6Li$, $^7Li$, and $^{23}Na$ were 58.89, 155.53 and 105.86 MHz, respectively. All spectra were acquired by using one-pulse program and were referenced to 1 M LiCl ($^6Li$ and $^7Li$) and 1 M NaCl ($^{23}Na$) solutions with chemical shifts at 0 ppm. The spinning rate $v_{rot}$ was set to 14 kHz. $^{23}Na$ spin-lattice relaxation times ($T_1$) were recorded by using the saturation recovery pulse sequence. The varying temperature experiments were protected by $N_2$ atmosphere. Nanoindentation measurement was taken on a nanoindentation tester (Agilent Nano Indenter G200) equipped with a three-sided pyramidal Berkovich diamond indenter. The applied standard loading, holding, and unloading times were 10, 5, and 10 s, respectively. During the testing, the load-displacement curves up to pellet cracking were recorded and utilized to calculate the Young's modulus $E$ and hardness $H$ using the Oliver–Pharr method. Indentations with maximum indentation load of 1 mN are conducted on the surface of SE pellets. The reduced modulus $E_r$ was determined by the unloading stiffness and projected contact area. By assuming a Poisson's ratio of 0.3 for samples and 0.07 for single crystalline diamond, their Young's modulus were estimated.

## Impedance spectrum test

For the conductivity measurement, silver was spread on both sides of the ceramic pellets as blocking electrodes. AC impedance spectra were recorded using a Solartron 1260 impedance analyzer over a frequency range of 5 MHz to 1 Hz, with an applied root mean square AC voltage of 30 mV. The temperature dependence of the conductivity was measured in the same way at several specific temperatures ranging from 25 to 175 °C. For conductivity test at each temperature, the samples were allowed to equilibrate for 2 h prior to measurements. The resistances of the Na|$Na_5SmSi_4O_{12}$|Na symmetric cells were tested under the same conditions.

## Electrochemical measurement

To obtain NVP cathode, NVP active material (70 wt.%), Super P conductive additive (20 wt.%) and carboxymethyl cellulose (CMC) binder (10 wt.%) were dissolved in water to form a homogeneous slurry, and then uniformly coated onto an aluminum foil current collector. After drying for 12 h at 80 °C, the electrode was punched into 1 cm diameter wafers for use with the loading mass of 1.0–1.5 mg cm$^{-2}$. Sodium foil was employed as anode. The $Na_5SmSi_4O_{12}$ pellet was used as both separator and electrolyte. 20 μL 1 M $NaClO_4$ in ethylene carbonate (EC) and propylene carbonate (PC) (1:1 v/v) with the addition of 5 vol.% fluoroethylene carbonate (FEC) was added as the interfacial wetting agent at the cathode side. The 2032-type coin cells were assembled in an argon-filled glovebox. Galvanostatic charge-discharge tests were performed in a cutoff potential window of 2.3–3.9 V by using Land-2100 automatic battery tester. All-solid-state Na|$Na_5SmSi_4O_{12}$|Na and Li|$Na_5SmSi_4O_{12}$|Li symmetric cells were assembled to test the sodium/lithium metal stripping/platting at 25 and 50 °C, respectively. In addition, electrochemical stable window of electrolytes was examined by cyclic voltammetry (CV) measurement with the stainless steel (SS)|$Na_5SmSi_4O_{12}$|Na cell in the voltage range of −1 to 8 V at a scanning rate of 5 mV s$^{-1}$. The direct current (DC) polarization measurement was performed on Ag|$Na_5SmSi_4O_{12}$|Ag cell with a 300 mV potential and the current response was measured for 100 min at ambient temperature. The CV measurement and the DC measurement were performed using a Bio-Logic electrochemical workstation.

## Computational methods

Density functional theory calculations were carried out using the VASP6.3[52,53] package following the setup we used in previous work[54–56]. Briefly, the PBE exchange–correlation functional was adopted with a planewave basis and a cutoff energy of 520 eV. The reciprocal space was sampled using Monkhorst–Pack grids with a spacing of 0.04 Å$^{-1}$. The convergence for electron self-consistent computations and structural optimizations are set to $10^{-4}$ eV atom$^{-1}$ and $10^{-3}$ eV atom$^{-1}$, respectively. Due to the need of large-scale molecular dynamics simulations, we trained a machine learning forcefield based on ab initio molecular dynamics simulations (AIMD)[54,55]. Trajectories from smaller systems with 5 compositions sampled evenly between $Na_5SmSi_4O_{12}$ and $Li_5SmSi_4O_{12}$ were collected at high temperatures together with their relaxation trajectories. The energy and forces were used as the label to train the model. The production MD simulations were carried out using such a forcefield. The systems were equilibrated at 600 K for 100 ps and gradually dropped to 300 K with a period of another 100 ps under an NPT thermostat at ambient pressure. Then an NVT production run was carried out at 300 K for 200 ps. The time step was 1 fs. For the interface model, we adopted a universal machine learning model which can capture the ground state structures and their energy. Interface models were built to mimic the interaction between the Li electrode and the crystalline and amorphous SE. The models were first equilibrated at 500 K for 1000 fs followed by energy minimization. The interfacial energy was calculated by subtracting the energy of the interfaces from the sum of energies of independent bulk phases.

## Data availability

The authors declare that the data generated in this study have been deposited in the Figshare database https://doi.org/10.6084/m9.figshare.24188418. Should any raw data files be needed in another format, they are available from the corresponding author upon reasonable request.

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

## Acknowledgements

F.D. acknowledges the support from the National Natural Science Foundation of China with Grant No. 12274176. M.T. acknowledges the support from the National Natural Science Foundation of China with Grant No. 21974007. F.D. also would like to thank the support from the Fundamental Research Funds for the Center Universities and Department of Science and Technology of Jilin Province with Grant Nos. 20220201118GX and 20210301021GX.

## Author contributions

F.D., M.T., and Z.L. designed and supervised the project. S.Y., B.Y., and W.J. designed the experiments to response the comments from the reviewers. G.S. performed materials synthesis, electrochemical tests, and wrote the manuscript. C.L. and M.T. performed NMR experiments and analyses. Z.L. carried out atomistic simulations and relevant analysis. Z.W., S.Y., G.C., and Z.S. revised the manuscript. All the authors participated in the discussion and provided constructive advice for the experimental design.

## Competing interests

The authors declare no competing interests.
