## [Peer Review File · Nature Communications]

REVIEWER COMMENTS

Reviewer #1 (Remarks to the Author):

In Na₅SmSi₄O₁₂-based ceramics, the authors have achieved an excellent critical current density and cycle properties in plating/stripping reactions against Na electrodes, together with a high conductivity of $3 \times 10^{-3} \text{ Scm}^{-2}$ at room temperature. An electrochemically induced amorphization is proposed as the origin for the excellent properties, and have been verified by theoretical calculations. The facts that the amorphization extends over the entire ~1mm-thick sample and that the total conductivity is mostly maintained after the amorphization are interesting new findings.

The novelty of the present study relies on this amorphization. Thus, sufficient experimental demonstration about the amorphization, its mechanism, and its contribution to the enhanced properties in the Na anode junction are supposed to be required.

(1) With regard to the amorphization, there is no doubt that the samples in the present study became at least partially amorphous. But it is not fully convinced, since quantitative information is lacking like: at what volume fraction, to where in a solid electrolyte sample and by what kinetics (e.g. as a function of the Na stripping/plating reaction) the amorphization is taking place. Such quantitative estimation is required, even partially.

The deduction of the mechanism of amorphization by experimental and theoretical collaboration in the Li system is interesting and convincing. However, since the authors have not attributed this knowledge to amorphization in the Na-SE interface, some conclusion is necessary, even speculatively. Why are the volume change and resultant strain induced in the Na-SE interface, as well? For example, does it allow excess Na to be inserted into the SE lattice near the Na interface, and does this induce a strain? If so, how does amorphization propagate into the interior of a sample? If it is not simply induced by an ionic current, does amorphization not occur, for example, in an NVP|SE|NVP cell?

The most basic and convenient way to quantify the weight fraction of amorphization would be to mix an internal standard (e.g. CeO₂) and perform Rietveld analysis. However, in this case, the sample is destroyed by crushing and only an averaged information is available.

(2) The conductivity of SE obtained by the authors, although notably high, is well foreseeable from the results presented in the pioneering work of Shannon et al. *Inorg. Chem.* (1978). This study should be properly cited and introduced (as in the authors' earlier paper) in the manuscript.

(3) Although the critical current density (CCD) is accepted as a measure of the 'goodness' of a SE and Na-SE interface, I consider not only the CCD but also the integrated current density per cycle, i.e., the critical charge density (unfortunately the abbreviation is the same), is important. In this respect, I agree that the authors have presented a well-defined integrated current density. I would like to know whether the critical integrated current density in this study is sufficiently high as compared to other similar studies to date.

(4) In the previous paper (Sun et al., *Energy Storage Mater.* 2021), the authors introduced an organic electrolyte into the NVP cathode, as in the present study, and it appears to have been titled "quasi"-solid-state battery. I think that removing "all-solid-state" from the original title may work well here.

(5) Minor points.

Line 4, page 12: "etc." has two citation numbers. It would be preferable to describe the materials.

The words of "ultarastable" or "ultraconformal" are somewhat exaggerated for a scientific paper.

Cell notation should be, for example, $\text{Na}|\text{Na}_5\text{SmSi}_4\text{O}_{12}|\text{Na}_3\text{V}_2(\text{PO}_4)_3$. The phase boundary is represented by a single line (not a double line as there is no salt bridge, etc.), and an anode is on the left side.

There are several values where the significant figures are too large. For example, $a = b = 22.14609 \text{ \AA}$ (7 digits may require a temperature control of 0.01 K level), $2.90 \times 10^{-3} \text{ S cm}^{-1}$ (requires very precise measurement of sample and electrode dimensions).

The crystalline system of space group R-3c is rhombohedral, not hexagonal; I understand that the lattice constants are in a hexagonal "setting". "Hexagonal" is found in the text and in Supplemental Table 2. In addition, "Y1" is supposed to be "Sm1".

Page 11, line 19: "holes" may be changed to "pores".

Reviewer #2 (Remarks to the Author):

The authors presented an interesting paper titled “Electrochemically induced crystalline-to-amorphization transformation in sodium samarium silicate solid electrolyte for long-lasting all-solid-state sodium metal batteries”. This paper suggests the advantages of $\text{Na}_5\text{SmSi}_4\text{O}_{12}$ solid electrolyte for long-life all-solid-state sodium metal batteries. However, due to the following reasons, I believe that Nature Communications cannot accept it as it is.

The authors claim to report a new member of the $\text{Na}_5\text{MSi}_4\text{O}_{12}$ family with $M=\text{Sm}$. However, solid electrolytes of this composition have been reported for a long time. For example, the following have been previously reported.

1. Solid State Ionics, Volumes 86–88, Part 1, July 1996, Pages 511-516

Synthesis and conduction properties of Na^+ superionic conductors of sodium samarium silicophosphates

2. Journal of the European Ceramic Society, Volume 26, Issues 4–5, 2006, Pages 619-622

Superionic conducting $\text{Na}_5\text{SmSi}_4\text{O}_{12}$ -type glass-ceramics: Crystallization condition and ionic conductivity

3. Journal of Electroceramics, Volume 24, 2010, Pages 83–90

Na^+ -fast ionic conducting glass-ceramics of silicophosphates

4. Solid State Ionics, Volume 262, 1 September 2014, Pages 604-608

Synthesis and Na^+ conduction properties of Nasicon-type glass–ceramics in the system $\text{Na}_2\text{O}-\text{Y}_2\text{O}_3-\text{R}_2\text{O}_3-\text{P}_2\text{O}_5-\text{SiO}_2$ (R = rare earth) and effect of Y substitution

5. Solid State Ionics, Volume 285, February 2016, Pages 143-154

Na^+ superionic conducting silicophosphate glass-ceramics – Review

6. Materials, 15, 2022, 1104. <https://doi.org/10.3390/ma15031104>

Influence of $R=\text{Y}, \text{Gd}, \text{Sm}$ on Crystallization and Sodium Ion Conductivity of $\text{Na}_5\text{RSi}_4\text{O}_{12}$ Phase

I feel that there is a lack of data on the subject of long life as a solid-state battery. The changes at the electrode-electrolyte interface after repeated charging and discharging of the battery over a long period of time and the results of analysis near the surface of the solid electrolyte should be presented.

The mechanism of conduction has also been reported since the 1980s, and the authors' previous report " $\text{Na}_{5}\text{YSi}_{4}\text{O}_{12}$: A sodium superionic conductor for ultrastable quasi solid-state sodium-ion batteries. batteries. Energy Storage Mater. 41, 196-202 (2021)" also discusses the same issues as in the present study.

The other figures are considered unnecessary as they are not very relevant and are not beyond the scope of the previous report.

Response to the Referees for Nature Communication

We would like to express our sincere gratitude to the editor and reviewers for their valuable contributions towards this report. We are truly grateful for the opportunity to address and clarify certain issues of interest in the revised manuscript. With this in mind, we have carefully reviewed and responded to each of the referee's comments point-to-point. All revisions have been highlighted in red in the uploaded manuscript.

Reviewer #1 (Remarks to the Author):

In Na₅SmSi₄O₁₂-based ceramics, the authors have achieved an excellent critical current density and cycle properties in plating/stripping reactions against Na electrodes, together with a high conductivity of $3 \times 10^{-3} \text{ S cm}^{-1}$ at room temperature. An electrochemically induced amorphization is proposed as the origin for the excellent properties, and have been verified by theoretical calculations. The facts that the amorphization extends over the entire ~1 mm-thick sample and that the total conductivity is mostly maintained after the amorphization are interesting new findings. The novelty of the present study relies on this amorphization. Thus, sufficient experimental demonstration about the amorphization, its mechanism, and its contribution to the enhanced properties in the Na anode junction are supposed to be required.

Author's Response: We would like to extend our sincere gratitude to the reviewer for his/her positive evaluation of our manuscript and invaluable suggestions. These suggestions play a crucial role in enhancing the overall quality of our work. We have diligently incorporated your feedback into our revision process, and we strongly believe that our detailed responses have addressed any potential concerns you may have had.

(1) With regard to the amorphization, there is no doubt that the samples in the present study became at least partially amorphous. But it is not fully convinced, since quantitative information is lacking like: at what volume fraction, to where in a solid electrolyte sample and by what kinetics (e.g. as a function of the Na stripping/plating

reaction) the amorphization is taking place. Such quantitative estimation is required, even partially.

Author's Response: Thank you for the kind advice. To confirm the crystalline-to-amorphous (CTA) transformation of $\text{Na}_5\text{SmSi}_4\text{O}_{12}$, X-ray diffraction was utilized to monitor the structural changes at various polishing depths, as depicted in Figure R1a. The analysis reveals that the solid electrolyte (SE) pellet, after undergoing 100 hours of plating/stripping at 0.15 mA cm^{-2} , exhibits a lack of reflections, which suggests complete amorphization near the surface of the $\text{Na}_5\text{SmSi}_4\text{O}_{12}$ SE. As the polishing depths increased, a limited number of weak reflections of $\text{Na}_5\text{SmSi}_4\text{O}_{12}$ emerge, indicative of a gradual amorphization process from the surface to the bulk. Upon polishing to a depth of 0.15 mm, sharp reflections are observed, suggesting that the crystalline SE may constitute the majority of the volumetric ratio within the entire $\text{Na}_5\text{SmSi}_4\text{O}_{12}$ pellet.

Figure R1. (a) Schematic and (b) XRD patterns of the $\text{Na}_5\text{SmSi}_4\text{O}_{12}$ polished to different depths after cycling 100 h at 0.15 mA cm^{-2} .

According to the reviewer's suggestion, we have employed the internal standard method to quantitatively assess the volume fraction of amorphous $\text{Na}_5\text{SmSi}_4\text{O}_{12}$. This involves mixing CeO_2 with the $\text{Na}_5\text{SmSi}_4\text{O}_{12}$ solid electrolyte (SE) at different plating/stripping stages. Figure R2 illustrates that the weight fraction of amorphization is directly proportional to both the cycling time (h) and current density (mA cm^{-2}). This correlation aligns with the time-resolved XRD results presented in Figure R1. It is observed that an increase in cycling time leads to a higher weight fraction of the amorphous phase. Similarly, at the same cycling time, a higher applied current density accelerates the amorphization process of the solid electrolyte. In the revised manuscript,

we have discussed this phenomenon as “To further assess the amorphous depth of $\text{Na}_5\text{SmSi}_4\text{O}_{12}$, XRD was utilized to monitor the structural changes at various polishing depths, as depicted in Supplementary Fig. 10. The analysis reveals that the SE pellet, after undergoing 100 hours of cycling at 0.15 mA cm^{-2} , exhibits no reflections, indicating complete amorphization near the surface of the $\text{Na}_5\text{SmSi}_4\text{O}_{12}$ SE. As the polishing depths increased, a limited number of weak reflections of $\text{Na}_5\text{SmSi}_4\text{O}_{12}$ emerge, which suggests a gradual amorphization process from the surface to the bulk. Upon polishing to a depth of 0.15 mm, sharp reflections appear which means the crystalline SE may constitute the majority of the volumetric ratio within the entire $\text{Na}_5\text{SmSi}_4\text{O}_{12}$ chip. To quantify the weight fraction of amorphization in an SE sample and by what kinetics, the internal standard method was employed to further assess the CTA transformation process, by mixing CeO_2 with the $\text{Na}_5\text{SmSi}_4\text{O}_{12}$ SE at different plating/stripping stages. The Rietveld refinements and the corresponding results are summarized in Supplementary Fig. 11 and Fig. 12, respectively, which clearly reveal the weight fraction of amorphization is strongly related to both the cycling time and current density. With increasing the cycling time, a higher weight fraction of the amorphous phase can be realized and a higher applied current density can accelerate the amorphization process.” from line 6 to 22 on Page 11, and line 1 on Page 12.

Figure R2. The amorphization weight fraction of $\text{Na}_5\text{SmSi}_4\text{O}_{12}$ determined by the internal standard method for different (a) cycling times and (b) current densities.

The deduction of the mechanism of amorphization by experimental and theoretical collaboration in the Li system is interesting and convincing. However, since the authors

have not attributed this knowledge to amorphization in the Na-SE interface, some conclusion is necessary, even speculatively. Why are the volume change and resultant strain induced in the Na-SE interface, as well? For example, does it allow excess Na to be inserted into the SE lattice near the Na interface, and does this induce a strain? If so, how does amorphization propagate into the interior of a sample? If it is not simply induced by an ionic current, does amorphization not occur, for example, in an NVP/SE/NVP cell?

Author's Response: We thank the reviewer for the insightful suggestion. In the crystal structure of $\text{Na}_5\text{SmSi}_4\text{O}_{12}$ (Figure R3a), vacancies at Na4, Na5 and Na6 sites are observed, which may facilitate the insertion of additional Na^+ ions. Therefore, $\text{Na}_5\text{SmSi}_4\text{O}_{12}$ is investigated as an anode material to validate this possible insertion. The discharge capacity of $\text{Na}_5\text{SmSi}_4\text{O}_{12}$ is demonstrated in Figure R3b. Such result is without the contribution from Super P, indicating that the SE possesses additional sites capable of accommodating Na^+ ions. Furthermore, EDS mapping was employed to monitor the changes in element ratios before and after cycling. A comparison of the data in Table R1 reveals an increase in Na content, supporting the speculation that the insertion of Na^+ ions into the vacancies leads to changes in size effect, resulting in lattice stress. As stress accumulates, the internal strain field leads to a gradual transformation of the material into an amorphous state. In the revised manuscript, we have discussed this phenomenon as “*As presented in the schematic crystal structure of $\text{Na}_5\text{SmSi}_4\text{O}_{12}$ (Supplementary Fig. 21a), vacancies are observed at Na4, Na5 and Na6 sites, which may accommodate additional Na ions' insertion and thus induce local variance. This possible insertion can be confirmed by the presence of the initial discharge capacity of $\text{Na}_5\text{SmSi}_4\text{O}_{12}$ (Supplementary Fig. 21b) and an increase in the Na content after cycling (EDS mapping, Supplementary Table 8). As the local stress accumulates, the internal strain field leads to the CTA transformation.*” between line 4 and 10, page 19.

Figure R3. (a) Schematical crystal structure of $\text{Na}_5\text{SmSi}_4\text{O}_{12}$; The initial charge-discharge profile of (b) $\text{Na}_5\text{SmSi}_4\text{O}_{12}$:Super P (7:2) and (c) Super P as the anode in the Na metal half cell.

Table R1. The comparison of element ratios of $\text{Na}_5\text{SmSi}_4\text{O}_{12}$ before and after cycling.

	Na	Sm	Si	O
Before cycling	4.9	1.0	4.2	12.1
After cycling	6.1	1.0	4.1	12.2

The most basic and convenient way to quantify the weight fraction of amorphization would be to mix an internal standard (e.g. CeO_2) and perform Rietveld analysis. However, in this case, the sample is destroyed by crushing and only an averaged information is available.

Author's Response: Thank you for the insightful advice. According to the reviewer's suggestion, we performed the internal standard method to evaluate the relationship between the weight fraction of amorphization and Na deposition/stripping, and the results are displayed in **Figure R2**. It can be seen that an increase in cycling time leads to a higher weight fraction of the amorphous phase. Similarly, at the same cycling time, a higher applied current density accelerates the amorphization process of the solid electrolyte. In the revised manuscript, we have added discussions as "To quantify the weight fraction of amorphization in an SE sample and by what kinetics, the internal standard method was employed to further assess the CTA transformation process, by mixing CeO_2 with the $\text{Na}_5\text{SmSi}_4\text{O}_{12}$ SE at different plating/stripping stages. The Rietveld refinements and the corresponding results are summarized in Supplementary Fig. 11

and Fig. 12, respectively, which clearly reveal the weight fraction of amorphization is strongly related to both the cycling time and current density. With increasing the cycling time, a higher weight fraction of the amorphous phase can be realized and a higher applied current density can accelerate the amorphization process.” between line 14 and 22 on Page 11 and line 1 on Page 12.

(2) The conductivity of SE obtained by the authors, although notably high, is well foreseeable from the results presented in the pioneering work of Shannon et al. Inorg. Chem. (1978). This study should be properly cited and introduced (as in the authors' earlier paper) in the manuscript.

Author's Response: Thank you for the suggestion. In the revised manuscript, we have cited Shannon et al.'s work and revised the manuscript accordingly as “In 1978, Shannon et al. first reported the synthesis of a new inorganic material $\text{Na}_5\text{MSi}_4\text{O}_{12}$ and found that the ionic conductivity gradually increases with the increase of M^{3+} ionic radius with ionic conductivity of $10^{-1} \text{ S cm}^{-1}$ at 200°C^{16} .” in page 3.

(3) Although the critical current density (CCD) is accepted as a measure of the 'goodness' of a SE and Na-SE interface, I consider not only the CCD but also the integrated current density per cycle, i.e., the critical charge density (unfortunately the abbreviation is the same), is important. In this respect, I agree that the authors have presented a well-defined integrated current density. I would like to know whether the critical integrated current density in this study is sufficiently high as compared to other similar studies to date.

Author's Response: Based on the reviewer's suggestions, we conducted a comparison between our reported critical integrated current density and the values obtained by other researchers^[1-11], as presented in Table R2. By utilizing the amorphous $\text{Na}_5\text{SmSi}_4\text{O}_{12}$, we are able to achieve a significantly enhanced critical current density (CCD) of 1.4 mA cm^{-2} . This improvement is in stark contrast to the lower initial crystalline stage CCD of 0.4 mA cm^{-2} . Our findings demonstrate that the amorphous $\text{Na}_5\text{SmSi}_4\text{O}_{12}$ enables a more intimate contact between the solid electrolyte (SE) and Na metal,

resulting in superior performance compared to oxide SEs that lacks interfacial modification with SE/Na metal anodes. In the revised manuscript, we have also added discussion as “As displayed in Fig. 3a and Supplementary Fig. 16, the amorphous $\text{Na}_5\text{SmSi}_4\text{O}_{12}$ facilitates intimate contact of SE with Na metal and brings essentially improved critical current density (CCD) of 1.4 mA cm^{-2} in comparison with the initial crystalline stage (0.4 mA cm^{-2}). This behavior is mainly because the uneven metal deposition leads to rapid growth of sodium dendrites along the crystalline $\text{Na}_5\text{SmSi}_4\text{O}_{12}$ grain boundary with a large deposition current and hence results in the short circuit rapidly. The result is also much higher than most reported values based on oxide SEs without and with SE/Na metal anode interfacial modification (Supplementary Table 6), indicating the superiority of amorphous SE interfaces.” between line 11 and 15 on Page 13 and line 1 to 4 on Page 14.

Table R2. A survey of critical current density of Na-based oxide solid-state electrolytes.

Electrolytes	Operating temperature	Critical current density	Deposited capacity	Ref.
amorphous $\text{Na}_5\text{SmSi}_4\text{O}_{12}$	25 °C	1.4 mA cm^{-2}	1.4 mA h cm^{-2}	This work
crystalline $\text{Na}_5\text{SmSi}_4\text{O}_{12}$	25 °C	0.4 mA cm^{-2}	0.4 mA h cm^{-2}	[1]
$\text{Na}_{4.9}\text{Sm}_{0.3}\text{Y}_{0.2}\text{Gd}_{0.2}\text{La}_{0.1}\text{Al}_{0.1}\text{Zr}_{0.1}\text{Si}_4\text{O}_{12}$	25 °C	0.6 mA cm^{-2}	0.6 mA cm^{-2}	[1]
$\text{Na}_{3.2}\text{Zr}_{1.9}\text{Mg}_{0.1}\text{Si}_2\text{PO}_{12}$	25 °C	0.5 mA cm^{-2}	$0.08 \text{ mA h cm}^{-2}$	[2]
$\text{AlF}_3\text{-Na}_3\text{Zr}_2\text{Si}_2\text{PO}_{12}$	60 °C	1.2 mA cm^{-2}	2.4 mA h cm^{-2}	[3]
$\text{Na}_3\text{Zr}_2\text{Si}_2\text{PO}_{12}\text{-10 wt.\% Na}_2\text{B}_4\text{O}_7$	25 °C	0.55 mA cm^{-2}	$0.275 \text{ mA h cm}^{-2}$	[4]
$\text{Na}_3\text{Zr}_2\text{Si}_2\text{PO}_{12}$	60 °C	0.6 mA cm^{-2}	0.5 mA h cm^{-2}	[5]
$\text{Na}_{3.4}\text{Mg}_{0.1}\text{Zr}_{1.9}\text{Si}_{2.2}\text{P}_{0.8}\text{O}_{12}$	60 °C	2.0 mA cm^{-2}	0.5 mA h cm^{-2}	[5]
$\text{Na}_{3.2}\text{Hf}_{1.9}\text{Ca}_{0.1}\text{Si}_2\text{PO}_{12}@ \text{SnO}_2$	60 °C	1.9 mA cm^{-2}	$0.475 \text{ mA h cm}^{-2}$	[6]
$\text{Na}_{3.2}\text{Hf}_{1.9}\text{Ca}_{0.1}\text{Si}_2\text{PO}_{12}@ \text{SnO}_2$	25 °C	1.2 mA cm^{-2}	0.3 mA h cm^{-2}	[6]
$\text{Na}_3\text{Zr}_2\text{Si}_2\text{PO}_{12}$	25 °C	0.4 mA cm^{-2}	0.4 mA h cm^{-2}	[7]
$\text{Na}_{3.2}\text{Hf}_{1.9}\text{Ca}_{0.1}\text{Si}_2\text{PO}_{12}\text{-CuO}$	25 °C	0.6 mA cm^{-2}	0.6 mA cm^{-2}	[8]
$\text{Na}_3\text{Zr}_2\text{Si}_2\text{PO}_{12}$	25 °C	0.15 mA cm^{-2}	$0.15 \text{ mA h cm}^{-2}$	[8]

$\text{Na}_3\text{Zr}_2\text{Si}_2\text{PO}_{12}$	25 °C	0.1 mA cm ⁻²	0.017 mA h cm ⁻²	[9]
$\text{Na}_3\text{Zr}_2\text{Si}_2\text{PO}_{12}\text{-SnO}_x/\text{Sn}$	25 °C	1.0 mA cm ⁻²	0.17 mA h cm ⁻²	[9]
$\text{Na}_3\text{Zr}_2\text{Si}_2\text{PO}_{12}\text{-TiO}_2$	25 °C	1.0 mA cm ⁻²	0.08 mA h cm ⁻²	[10]
$\text{Na}_3\text{Zr}_2\text{Si}_2\text{PO}_{12}$	25~27 °C	0.6 mA cm ⁻²	0.3 mA h cm ⁻²	[11]

references

- [1] Sun G, *et al.* High-Entropy Solid-State Na-Ion Conductor for Stable Sodium-Metal Batteries. *Chem. Eur. J.* **29**, e202300413 (2023).
- [2] Fu H, *et al.* Reducing Interfacial Resistance by Na-SiO₂ Composite Anode for NASICON-Based Solid-State Sodium Battery. *ACS Mater. Lett.* **2**, 127-132 (2019).
- [3] Miao X, *et al.* AlF₃-modified anode-electrolyte interface for effective Na dendrites restriction in NASICON-based solid-state electrolyte. *Energy Storage Mater.* **30**, 170-178 (2020).
- [4] Zhao Y, Wang C, Dai Y, Jin H. Homogeneous Na⁺ transfer dynamic at Na/Na₃Zr₂Si₂PO₁₂ interface for all solid-state sodium metal batteries. *Nano Energy* **88**, 106293 (2021).
- [5] Shen L, Yang J, Liu G, Avdeev M, Yao X. High ionic conductivity and dendrite-resistant NASICON solid electrolyte for all-solid-state sodium batteries. *Mater. Today Energy* **20**, 100691 (2021).
- [6] Tian H, Liu S, Deng L, Wang L, Dai L. New-type Hf-based NASICON electrolyte for solid-state Na-ion batteries with superior long-cycling stability and rate capability. *Energy Storage Mater.* **39**, 232-238 (2021).
- [7] Wang C, Jin H, Zhao Y. Surface Potential Regulation Realizing Stable Sodium/Na₃Zr₂Si₂PO₁₂ Interface for Room-Temperature Sodium Metal Batteries. *Small* **17**, e2100974 (2021).
- [8] Sun Z, *et al.* Active Control of Interface Dynamics in NASICON-Based Rechargeable Solid-State Sodium Batteries. *Nano Lett.* **22**, 7187-7194 (2022).
- [9] Yang J, *et al.* Improving Na/Na₃Zr₂Si₂PO₁₂ Interface via SnO_x/Sn Film for High-Performance Solid-State Sodium Metal Batteries. *Small Methods* **5**, e2100339

(2021).

- [10] Gao Z, *et al.* TiO₂ as Second Phase in Na₃Zr₂Si₂PO₁₂ to Suppress Dendrite Growth in Sodium Metal Solid-State Batteries. *Adv. Energy Mater.* **12**, 2103607 (2022).
- [11] Wang X, Chen J, Wang D, Mao Z. Improving the alkali metal electrode/inorganic solid electrolyte contact via room-temperature ultrasound solid welding. *Nat. Commun.* **12**, 7109 (2021).

(4) In the previous paper (Sun et al., Energy Storage Mater. 2021), the authors introduced an organic electrolyte into the NVP cathode, as in the present study, and it appears to have been titled "quasi"-solid-state battery. I think that removing "all-solid-state" from the original title may work well here.

Author's Response: In accordance with the reviewer's advice, we have made a modification throughout the manuscript, revising the term "all-solid-state batteries" into "quasi-solid-state batteries".

(5) Minor points.

Line 4, page 12: "etc." has two citation numbers. It would be preferable to describe the materials.

The words of "ultrarastable" or "ultraconformal" are somewhat exaggerated for a scientific paper.

Cell notation should be, for example, Na|Na₅SmSi₄O₁₂/Na₃V₂(PO₄)₃. The phase boundary is represented by a single line (not a double line as there is no salt bridge, etc.), and an anode is on the left side.

There are several values where the significant figures are too large. For example, $a = b = 22.14609$ Å (7 digits may require a temperature control of 0.01 K level), 2.90×10^{-3} S cm⁻¹ (requires very precise measurement of sample and electrode dimensions).

The crystalline system of space group R-3c is rhombohedral, not hexagonal; I understand that the lattice constants are in a hexagonal "setting". "Hexagonal" is found in the text and in Supplemental Table 2. In addition, "Y1" is supposed to be "Sm1".

Page 11, line 19: "holes" may be changed to "pores".

Authors' Response: We sincerely appreciate the valuable advice provided by the reviewer, which has significantly enhanced the accuracy and clarity of our manuscript. We have diligently addressed all the previously identified errors, appropriately highlighted in red within the revised manuscript.

	Original content	Revised content	Location
1	etc ^{28, 29}	AlF ₃ ²⁸ and polymer with intrinsic nanoporosity ²⁹	Page 4
2	ultrastable	stable	Page 5
3	ultraconformal	compact	Page 11, 17
4	Na ₃ V ₂ (PO ₄) ₃ Na ₅ SmSi ₄ O ₁₂ Na	Na Na ₅ SmSi ₄ O ₁₂ Na ₃ V ₂ (PO ₄) ₃	in full text
5	Na Na ₅ SmSi ₄ O ₁₂ Na	Na Na ₅ SmSi ₄ O ₁₂ Na	in full text
6	Li Na ₅ SmSi ₄ O ₁₂ Li	Li Na ₅ SmSi ₄ O ₁₂ Li	in full text
7	$a = b = 22.14609 \text{ \AA}$, $c = 12.68858 \text{ \AA}$	$a = b = 22.15 \text{ \AA}$, $c = 12.69 \text{ \AA}$	Page 6
8	$2.90 \times 10^{-3} \text{ S cm}^{-1}$	$2.9 \times 10^{-3} \text{ S cm}^{-1}$	in full text
9	hexagonal	rhombohedral	Page 5, 6, 21
10	Y1	Sm1	Page 33 in SI
11	holes	pores	Page 10

Reviewer #2 (Remarks to the Author):

The authors presented an interesting paper titled “Electrochemically induced crystalline-to-amorphization transformation in sodium samarium silicate solid electrolyte for long-lasting all-solid-state sodium metal batteries”. This paper suggests the advantages of $\text{Na}_5\text{SmSi}_4\text{O}_{12}$ solid electrolyte for long-life all-solid-state sodium metal batteries. However, due to the following reasons, I believe that Nature Communications cannot accept it as it is.

The authors claim to report a new member of the $\text{Na}_5\text{MSi}_4\text{O}_{12}$ family with $M=\text{Sm}$. However, solid electrolytes of this composition have been reported for a long time. For example, the following have been previously reported.

1. Solid State Ionics, Volumes 86–88, Part 1, July 1996, Pages 511-516

Synthesis and conduction properties of Na^+ superionic conductors of sodium samarium silicophosphates

2. Journal of the European Ceramic Society, Volume 26, Issues 4–5, 2006, Pages 619-622

Superionic conducting $\text{Na}_5\text{SmSi}_4\text{O}_{12}$ -type glass-ceramics: Crystallization condition and ionic conductivity

3. Journal of Electroceramics, Volume 24, 2010, Pages 83–90

Na^+ -fast ionic conducting glass-ceramics of silicophosphates

4. Solid State Ionics, Volume 262, 1 September 2014, Pages 604-608

Synthesis and Na^+ conduction properties of Nasicon-type glass–ceramics in the system $\text{Na}_2\text{O}-\text{Y}_2\text{O}_3-\text{R}_2\text{O}_3-\text{P}_2\text{O}_5-\text{SiO}_2$ ($R = \text{rare earth}$) and effect of Y substitution

5. Solid State Ionics, Volume 285, February 2016, Pages 143-154

Na^+ superionic conducting silicophosphate glass-ceramics – Review

6. Materials, 15, 2022, 1104. <https://doi.org/10.3390/ma15031104>

Influence of $R=\text{Y}, \text{Gd}, \text{Sm}$ on Crystallization and Sodium Ion Conductivity of $\text{Na}_5\text{RSi}_4\text{O}_{12}$ Phase

Author’s Response: Thank you for the insightful comment and we acknowledge that $\text{Na}_5\text{MSi}_4\text{O}_{12}$ is not a new family of ionic conductors. However, earlier research has primarily focused on reporting the ionic conductivity of these materials at high

temperatures, while what we reported here focuses on the crystalline-to-amorphous (CTA) transition and its stabilization of the interface for solid-state batteries. Accordingly, we have modified our statements. It is worthwhile to note that, to meet the eager demand for low-cost and high-safety batteries, solid-state sodium metal batteries appear as promising devices for future energy storage systems. So far, the reported Na-based SE is limited to Na- β' / β'' -Al₂O₃, NASICON-type materials, Na₃PS₄ system, Na₁₁Sn₂PS₁₂ system, etc. Na₅MSi₄O₁₂ is a promising type of Na⁺ ionic conductor with high ionic conductivity above 10⁻³ S cm⁻¹, while there are rare studies on their application in solid-state sodium metal batteries. In our earlier study in Energy Storage Material^[1], we report the synthesis and ionic transfer mechanism, which opens a new application area for such kind of material. In this manuscript, we not only realize a higher ionic conductivity in Na₅SmSi₄O₁₂ with a lower reaction temperature but also discover a CTA transition during Na⁺ plating/stripping processes, which may propose a mechanism to achieve a conformal interface between Na metal and SE in the solid-state battery assembly. These discoveries can further propel the application of Na₅MSi₄O₁₂ materials in solid-state batteries.

[1] Sun G, *et al.* Na₅YSi₄O₁₂: A sodium superionic conductor for ultrastable quasi-solid-state sodium-ion batteries. *Energy Storage Mater.* **41**, 196-202 (2021).

I feel that there is a lack of data on the subject of long life as a solid-state battery. The changes at the electrode-electrolyte interface after repeated charging and discharging of the battery over a long period of time and the results of analysis near the surface of the solid electrolyte should be presented.

Author's Response: According to the reviewer's suggestion, we have included the cycle lives of the state-of-the-art solid-state full cell in Table R3. It is obvious that our proposed Na|Na₅SmSi₄O₁₂|Na₃V₂(PO₄)₃ batteries demonstrate very comparative cycle lives in state-of-the-art solid-state batteries^[1-12]. In the revised Supplementary material, we have added this table as Supplementary Table 7 and discussed it accordingly.

Table R3. A survey of cycle performance of oxide electrolytes-based solid-state cells.

Solid-state cells description	Operating temperature	Cycle performance	Ref.
Na Na₅SmSi₄O₁₂ Na₃V₂(PO₄)₃	25 °C	4000 cycles 100%	This work
Na Na ₅ YSi ₄ O ₁₂ Na ₃ V ₂ (PO ₄) ₃	25 °C	500 cycles 100%	[1]
Na/β"-Al ₂ O ₃ /NaTi ₂ (PO ₄) ₃	25 °C	50 cycles 75.1%	[2]
Na/β"-Al ₂ O ₃ /Na _{0.66} Ni _{0.33} Mn _{0.67} O ₂	70 °C	10000 cycles 90%	[3]
Na/Na ₃ Zr ₂ (Si ₂ PO ₁₂)/NVP	50 °C	100 cycles 98%	[4]
Na/Na ₃ Zr ₂ Si ₂ PO ₁₂ /Na ₂ MnFe(CN) ₆	60 °C	200 cycles 89.2%	[5]
Na/beta-alumina/PTO	60 °C	50 cycles 80%	[6]
Na/Na _{3.2} Zr _{1.8} Ca _{0.1} Si ₂ PO ₁₂ /NVP	25 °C	500 cycles 98%	[7]
Na/Na _{3.4} Zr _{1.8} Mg _{0.2} PO ₁₂ /NaCrO ₂	25 °C	1755 cycles 87%	[8]
Na/polydopamine-Na _{3.4} Zr _{1.9} Zn _{0.1} Si _{2.2} P _{0.8} O ₁₂ /FeS ₂	60 °C	300 cycles 73.3%	[9]
UW-Na/Na ₃ Zr ₂ Si ₂ PO ₁₂ /NVP	25–27 °C	900 cycles 89.8%	[10]
Na/Na _{3.3} Zr _{1.7} La _{0.3} Si ₂ PO ₁₂ /IL/NVP	25 °C	10000 cycles 100%	[11]
Na/Na ₃ Zr ₂ Si ₂ PO ₁₂ /NVP	25 °C	100 cycles 97%	[12]

references

- [1] Sun G, *et al.* Na₅YSi₄O₁₂: A sodium superionic conductor for ultrastable quasi-solid-state sodium-ion batteries. *Energy Storage Mater.* **41**, 196-202 (2021).
- [2] Zhao K, *et al.* A room temperature solid-state rechargeable sodium ion cell based on a ceramic Na-β"-Al₂O₃ electrolyte and NaTi₂(PO₄)₃ cathode. *Electrochem. Commun.* **69**, 59-63 (2016).
- [3] Liu L, *et al.* Toothpaste-like Electrode: A Novel Approach to Optimize the Interface for Solid-State Sodium-Ion Batteries with Ultralong Cycle Life. *ACS Appl. Mater. Interfaces* **8**, 32631-32636 (2016).
- [4] Gao H, Xue L, Xin S, Park K, Goodenough JB. A Plastic-Crystal Electrolyte Interphase for All-Solid-State Sodium Batteries. *Angew. Chem. Int. Ed. Engl.* **56**, 5541-5545 (2017).
- [5] Gao H, Xin S, Xue L, Goodenough JB. Stabilizing a High-Energy-Density Rechargeable Sodium Battery with a Solid Electrolyte. *Chem* **4**, 833-844 (2018).

- [6] Chi X, *et al.* A high-energy quinone-based all-solid-state sodium metal battery. *Nano Energy* **62**, 718-724 (2019).
- [7] Lu Y, Alonso JA, Yi Q, Lu L, Wang ZL, Sun C. A High-Performance Monolithic Solid-State Sodium Battery with Ca²⁺ Doped Na₃Zr₂Si₂PO₁₂ Electrolyte. *Adv. Energy Mater.* **9**, 1901205 (2019).
- [8] Wang C, *et al.* Grain Boundary Design of Solid Electrolyte Actualizing Stable All-Solid-State Sodium Batteries. *Small* **17**, e2103819 (2021).
- [9] Yang J, *et al.* Ultrastable All-Solid-State Sodium Rechargeable Batteries. *ACS Energy Lett.* **5**, 2835-2841 (2020).
- [10] Wang X, Chen J, Wang D, Mao Z. Improving the alkali metal electrode/inorganic solid electrolyte contact via room-temperature ultrasound solid welding. *Nat. Commun.* **12**, 7109 (2021).
- [11] Zhang Z, *et al.* A Self-Forming Composite Electrolyte for Solid-State Sodium Battery with Ultralong Cycle Life. *Adv. Energy Mater.* **7**, 1601196 (2017).
- [12] Yang J, *et al.* Improving Na/Na₃Zr₂Si₂PO₁₂ Interface via SnO_x/Sn Film for High-Performance Solid-State Sodium Metal Batteries. *Small Methods* **5**, e2100339 (2021).

In addition, the changes in the electrode-electrolyte interface and SE surface were studied in detail. First, the Nyquist plots of the symmetric Na|Na₅SmSi₄O₁₂|Na cell were recorded after different cycling times to reveal the changes in the internal resistance (Fig. 2b and Supplementary Table 4). As the sodium plating/stripping proceeds, the resistance of SE (both R_b and R_{GB}) remains nearly unchanged, while the interfacial resistance (R_{int}) between sodium metal and electrolyte decreases at an initial few cycles and then stabilizes at a relatively low value, demonstrating good compatibility between Na₅SmSi₄O₁₂ and sodium metal. To reveal the interfacial morphology evolution, SEM images of the electrode-electrolyte interface at different plating/stripping stages (Fig. 2c and 2d) were further recorded, indicating the disappearance of interface gap between Na and Na₅SmSi₄O₁₂ SE with the result of increased contact areas after cycling.

Furthermore, XRD patterns of Na₅SmSi₄O₁₂ show a CTA transition after cycling

for 200 h (Fig. 2e). To further confirm this CTA transition, HRTEM and SAED measurements before and after cycling were recorded (Supplementary Fig. 9). There are no observed lattice fringes (Fig. 2f) and diffraction spots (Fig. 2g) for the cycled SE sample, confirming the CTA transition once more. Owing to the intrinsic characteristics of the amorphous materials, it is difficult to determine the possible local coordination based on XRD measurement. Therefore, Raman spectra were then used to examine the potential short-range vibration changes in the chemical bonding. As expected, the amorphous $\text{Na}_5\text{SmSi}_4\text{O}_{12}$ SE maintains nearly all the vibrational peaks without new vibrations confirming no chemical reaction between the interface of SE and Na metal (Supplementary Fig. 10). While the disappearance of the peak at 1041 cm^{-1} might be related to the damage of the fracture of Si-O bond by the amorphous transition. Furthermore, X-ray photoelectron spectroscopy (Supplementary Fig. 11) suggests that there is no change in the Sm 3d, Na 1s and Si 2p XPS spectra after cycling, indicative of no redox reaction occurred between sodium metal and $\text{Na}_5\text{SmSi}_4\text{O}_{12}$ once more.

Moreover, the improved properties of bulk materials and amorphous interface can be understood in terms of the following two aspects: On one hand, the ionic conductivity of bulk material was improved. Figure 4a shows the ^{23}Na NMR spectra before and after metallic Na cycling by using crystalline $\text{Na}_5\text{SmSi}_4\text{O}_{12}$ as an electrolyte. The ^{23}Na NMR spectrum did not change significantly before and after the Na cycling, indicating the stability of the structure. The changes for Na4, Na5, and Na6 are obvious, indicating their redistribution upon cycling. In addition, the activation energy of the mobile Na of the cycled $\text{Na}_5\text{SmSi}_4\text{O}_{12}$ is 0.07 eV (Fig. 4b), which is much lower than that of the pristine state (0.13 eV in Fig. 1k), indicating the amorphous interface and bulk materials greatly reduces the activation energy, further resulting in higher conductivity of crystalline $\text{Na}_5\text{SmSi}_4\text{O}_{12}$. On the other hand, the interfacial issues, such as high interfacial resistance and metal dendrite growth, are strongly alleviated. The amorphous $\text{Na}_5\text{SmSi}_4\text{O}_{12}$ could enhance the wettability with significantly reduced interface resistance by reducing the interface energy between Na and $\text{Na}_5\text{SmSi}_4\text{O}_{12}$ since amorphous $\text{Na}_5\text{SmSi}_4\text{O}_{12}$ exhibits lower interfacial energy of 0.33 J m^{-2} with sodium than crystalline $\text{Na}_5\text{SmSi}_4\text{O}_{12}$ (0.56 J m^{-2}) (Supplementary Fig. 16). In addition,

Young's modulus E and hardness H of the amorphous $\text{Na}_5\text{SmSi}_4\text{O}_{12}$ are calculated to be ~ 79.9 GPa and ~ 3.8 GPa, respectively, higher than the crystalline $\text{Na}_5\text{SmSi}_4\text{O}_{12}$ (~ 72.6 GPa and ~ 2.8 GPa), beneficial to inhibiting the dendrite growth.

The mechanism of conduction has also been reported since the 1980s, and the authors' previous report "Na₅YSi₄O₁₂: A sodium superionic conductor for ultrastable quasi solid-state sodium-ion batteries. batteries. Energy Storage Mater. 41, 196-202 (2021)" also discusses the same issues as in the present study.

The other figures are considered unnecessary as they are not very relevant and are not beyond the scope of the previous report.

Author's Response: Thank you for the reviewer's kind suggestion. Besides the reports on the synthesis and ionic conducting mechanism, the most interesting finding in our manuscript is the discovery of "crystalline-into-amorphous transition" in $\text{Na}_5\text{SmSi}_4\text{O}_{12}$ SE during Na plating/stripping, which leads to faster ionic transport and superior interfacial properties.

We fully agree with the review's comment and revise Figure 1 and the corresponding discussion part in the revised manuscript.

Fig. 1. Crystal structure and sodium-ion conduction characteristic of crystalline $\text{Na}_5\text{SmSi}_4\text{O}_{12}$. **a** Rietveld refinement based on the powder XRD. **b** Minimum potential

energy path along Na^+ diffusion route in crystalline $\text{Na}_5\text{SmSi}_4\text{O}_{12}$. **c** Solid-state ^{23}Na NMR spectrum and its simulation for the crystalline $\text{Na}_5\text{SmSi}_4\text{O}_{12}$. The gray line is experimental data and the green-dashed line is the sum of simulation. **d** Saturation recovery fitting curve for the data obtained at room temperature. **e** Temperature dependence of ^{23}Na NMR relaxation rate as a function of temperature in K^{-1} . The solid line is the fit according to Eq. (1). The derivation of the data is not used for the fit.

REVIEWERS' COMMENTS

Reviewer #1 (Remarks to the Author):

The authors responded well to the review comments, and the revised manuscript and study are considered more complete. I especially appreciate the deeper discussion on critical 'charge' density.

One small point, in response to my comment about the number of digits of lattice constants, the revised version employs four digits, $a = b = 22.15$, $c = 12.69$ Å, which I think is too few as a result of the analysis; it would be appropriate to report around 5 or 6 digits.

Reviewer #2 (Remarks to the Author):

Since you have responded to my comments, I think I can accept the manuscript of this paper.

Response to the Referees for Nature Communication

We would like to express our sincere gratitude for the decision to accept our manuscript after revision. We deeply appreciate the time and effort the editor and the reviewers have dedicated to reviewing and providing constructive feedback on our work. We've carefully addressed the suggested revisions, which are highlighted in red in the revised manuscript. We look forward to the final publication and hope that our work will contribute to the broader scientific community.

Reviewer #1 (Remarks to the Author):

The authors responded well to the review comments, and the revised manuscript and study are considered more complete. I especially appreciate the deeper discussion on critical 'charge' density.

One small point, in response to my comment about the number of digits of lattice constants, the revised version employs four digits, $a = b = 22.15$, $c = 12.69$ Å, which I think is too few as a result of the analysis; it would be appropriate to report around 5 or 6 digits.

Author's Response: We are very grateful for your valuable comments, which have been instrumental in refining our manuscript and enhancing its quality. According to your suggestion, we have kept 6 digits for the lattice constant and revised the manuscript as “*All the diffraction peaks can be indexed into a rhombohedral system with space group $R\bar{3}c$, and the lattice parameters are calculated as $a = b = 22.1461$ Å, $c = 12.6886$ Å.*” on page 6.

Reviewer #2 (Remarks to the Author):

Since you have responded to my comments, I think I can accept the manuscript of this paper.

Author's Response: Thank you very much for your positive feedback and for agreeing to accept our manuscript. We appreciate the time and effort you have invested in reviewing our work.